# Evaluating the Quality of Hallucination Benchmarks for Large Vision-Language Models

## Abstract

Despite the rapid progress and outstanding performance of Large Vision-Language Models (LVLMs) in recent years, LVLMs have been plagued by the issue of hallucination, i.e., LVLMs tend to generate responses that are inconsistent with the corresponding visual inputs. To evaluate the degree of hallucination in LVLMs, previous works have proposed a series of benchmarks featuring different types of tasks and evaluation metrics. However, we find that the quality of the existing hallucination benchmarks varies, with some suffering from problems, e.g., inconsistent evaluation results under repeated tests, and misalignment with human evaluation. To this end, we propose a **H**allucination benchmark **Q**uality **M**easurement framework (**HQM**), which leverages various indicators to assess the reliability and validity of existing hallucination benchmarks separately. Specifically, for reliability we explore test-retest reliability and parallel-forms reliability, while for validity we examine criterion validity and coverage of hallucination types. Furthermore, we construct a **H**igh-**Q**uality **H**allucination Benchmark (**HQH**) for LVLMs, which demonstrates superior reliability and validity under our HQM framework. We conduct an extensive evaluation of over 10 representative LVLMs, including GPT-4o and Gemini-1.5-Pro, to provide an in-depth analysis of the hallucination issues in existing models. Our benchmark is publicly available at https://github.com/HQHBench/HQHBench.

## 1 Introduction

In recent years, the rise of Large Language Models (LLMs) has led to a great revolution in the field of artificial intelligence. Building on the success of LLMs, Large Vision-Language Models (LVLMs), sometimes referred to as Large Multimodal Models (LMMs), have made remarkable advancements. These models usually use LLMs as the foundational architecture and align features from other modalities accordingly, demonstrating exceptional capabilities across various multimodal tasks, such as image captioning and visual question answering (VQA). Despite their outstanding performance, LVLMs are significantly plagued by the issue of hallucination, which could lead to harmful consequences, particularly when users without sufficient domain knowledge over-rely on the models.

The original concept of hallucination is introduced for LLMs and categorized into factuality hallucination and faithfulness hallucination (Huang et al., 2023; Ji et al., 2023; Rawte et al., 2023). Factuality hallucination occurs when the generated content is inconsistent with real-world facts, while faithfulness hallucination refers to the discrepancy between the generated content and the context provided by the input instruction or output content itself. Compared to LLMs, hallucination in LVLMs is defined as inconsistency of the generated textual content and the visual input (Bai et al., 2024; Liu et al., 2024), emphasizing the multimodal inconsistency.

To assess the degree of hallucination in LVLMs, previous studies have proposed a series of hallucination benchmarks, supporting evaluation of closed-ended tasks and open-ended tasks. Closed-ended tasks include yes-or-no questions and multiple-choice questions, while open-ended tasks contain image captioning and free-form VQA. However, we find that some benchmarks suffer from quality issues, such as inconsistent evaluation results under repeated tests, misalignment with human eval-

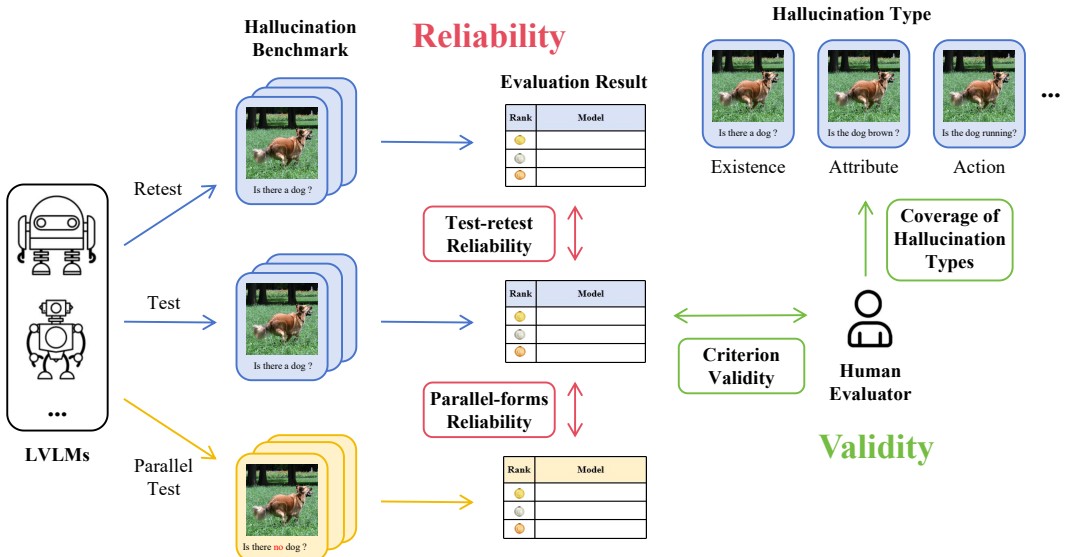

Figure 1: Overview of our Hallucination benchmark Quality Measurement framework (HQM), assessing both reliability and validity. For reliability, we explore test-retest reliability and parallel-forms reliability, examining whether the evaluation results are consistent under repeated tests and parallel tests. For validity, we measure criterion validity and the coverage of hallucination types, focusing on whether the benchmark evaluation is aligned with human evaluation and comprehensive.

uation, and limited coverage of hallucination types (Yifan Li & Wen, 2023; Lovenia et al., 2023; Ben-Kish et al., 2024), which raise doubts about the trustworthiness of their evaluation results. Thus, it is necessary to measure the quality of existing hallucination benchmarks.

Inspired by psychometrics (Furr, 2021; Raykov & Marcoulides, 2011; Rust & Golombok, 2014), we propose a framework of quality measurement for hallucination benchmarks from the perspective of reliability and validity. An overview of our quality measurement framework is illustrated in Figure 1. For reliability, we assess test-retest reliability and parallel-forms reliability, examining whether the evaluation results are consistent under repeated tests and parallel tests. For validity, we measure criterion validity, i.e., whether the evaluation results are aligned with human evaluation, and the coverage of hallucination types. Through detailed analysis, we summarize the strengths and limitations of existing benchmarks as follows. Firstly, we argue that benchmarks of closed-ended tasks offer efficient automated evaluation but exhibit certain deficiencies in reliability since LVLMs are susceptible to response bias (Tjuatja et al., 2023) introduced by task settings, such as acquiescence bias and dissent bias in yes-or-no questions (Yifan Li & Wen, 2023; Fu et al., 2023), position bias in multiple-choice questions (Zheng et al., 2024; Xu et al., 2023). Such bias manifests as the tendency to answer "*yes*" or "*no*" to yes-or-no questions and select a specific option in multiple-choice questions. In contrast, benchmarks of open-ended tasks avoid response bias by allowing more freedom in responses, but they primarily suffer from validity issues, with more severe misalignment between their evaluation and human evaluation.

Considering the balance between reliability and validity, we opt to build our hallucination benchmark on open-ended tasks, specifically free-form VQA. We collect images from the validation set of Visual Genome (Krishna et al., 2017) dataset and design image-instruction pairs covering comprehensive types of hallucination, including attribute, action, counting, environment, (spatial) relation, comparison, OCR, and existence. To ensure the data quality, we conduct a manual review of all image-instruction pairs and remove low-quality samples. As for metric, existing free-form VQA benchmarks use hallucination score, which leverages external LLMs like GPT (OpenAI, 2022) to assign a specific score to the hallucination level of model response. We think such scoring-based metrics are too difficult for current LLMs, resulting in inconsistent scores across repeated or parallel tests, as well as inaccurate scores that are misaligned with human evaluation. Instead, we employ a simplified process: given detailed image information, the model only needs to determine whether the response is hallucinated. Thus, the hallucination rate can be computed as the evaluation metric.

Compared to score-based metrics, our simplified process is more effective, which can minimize the gap in evaluation capabilities between GPT and human evaluators, enhancing the reliability and validity of our benchmark.

In conclusion, our contributions are as follows:

- We propose a **H**allucination benchmark **Q**uality **M**easurement framework (**HQM**) for LVLMs, which leverages different indicators to assess the reliability and validity.
- Under our proposed quality measurement framework, we construct a new **H**igh-**Q**uality **H**allucination Benchmark (**HQH**) with improved reliability and validity.
- To provide an in-depth analysis of the hallucination issues in existing models, we conduct a large-scale evaluation of over 10 representative LVLMs using our benchmark **HQH**, including GPT-4o (OpenAI, 2024) and Gemini-1.5-Pro (Team et al., 2023).

## 2 RELATED WORKS

### 2.1 LARGE VISION-LANGUAGE MODELS

Built on the success of LLMs, LVLMs have rapidly developed, demonstrating strong capabilities. Researchers have constructed a series of advanced LVLMs using various methods. For example, BLIP2 (Li et al., 2023b) adopts a lightweight Q-Former architecture and uses cross-attention mechanisms to align textual and visual representations. InstructBLIP (Dai et al., 2024) incorporates textual instructions into the Q-Former, enhancing the model performance. LLaVA (Liu et al., 2023b) is the first to introduce instruction tuning techniques to the multimodal field, forming the most mature open-source multimodal model. The emergence of other open-source models such as MiniGPT-4 (Zhu et al., 2023), Otter (Li et al., 2023a), Shikra (Chen et al., 2023), and Qwen-VL (Bai et al., 2023) have further propelled the development of LVLMs. Additionally, many powerful closed-source LVLMs, including Gemini-1.5-Pro (Team et al., 2023) and GPT-4o (OpenAI, 2024), have publicly released their APIs, promoting the development of downstream applications. In this paper, we use these open-source LVLMs as test models under our HQM framework, and benchmark them along with several closed-source models on our HQH.

### 2.2 HALLUCINATION BENCHMARKS FOR LVLMS

In the context of LVLMs, hallucination refers to the inconsistency of the generated textual content and the visual input (Bai et al., 2024; Liu et al., 2024). To evaluate the degree of hallucination in LVLMs, various hallucination benchmarks have been proposed, which can be divided into two categories, closed-ended tasks and open-ended tasks. For closed-ended tasks, previous works design yes-or-no questions or multiple-choice questions (Lu et al., 2023), using accuracy as evaluation metric. For example, POPE (Yifan Li & Wen, 2023) constructs yes-or-no questions based on different polling strategies to detect whether the responses contain non-existent objects. Following works like AMBER (Wang et al., 2023a) extend yes-or-no questions to other types of hallucination. HallusionBench (Guan et al., 2023) manually constructs yes-or-no pairs with an innovative structure by human experts, further measuring more fine-grained hallucination. For open-ended tasks, existing works often employ image captioning or free-form VQA. One kind of evaluation metric is CHAIR (Rohrbach et al., 2018) and its variants (Jing et al., 2023; Ben-Kish et al., 2024), which calculates the proportion of hallucinated objects to all objects mentioned in the response and is mostly used for image captioning. For instance, OpenCHAIR (Ben-Kish et al., 2024) leverages OCH, which expands CHAIR to an open vocabulary, to evaluate the hallucination in image descriptions. Another kind of metric hallucination score utilizes external LLMs like GPT (OpenAI, 2022) to grade the degree of hallucination and give exact scores to the generated responses, which is relatively more popular in free-form VQA benchmarks like MMHal (Sun et al., 2023) and GAVIE (Liu et al., 2023a). We select several representative benchmarks to conduct quality measurement.

## 3 HALLUCINATION BENCHMARK QUALITY MEASUREMENT FRAMEWORK

Inspired by psychometrics (Rust & Golombok, 2014; Raykov & Marcoulides, 2011; Furr, 2021), we propose a hallucination benchmark quality evaluation framework HQM. Psychometrics, i.e., the

science of psychological assessment, has been long utilized to measure the psychological construct of human (Rust & Golombok, 2014). To some degree, AI benchmarks for evaluating model capabilities have similarities with psychological tests used to assess human psychological constructs like intelligence. Therefore, the integration of psychometrics into AI evaluation has received increasing attention (Wang et al., 2023b; Pellert et al., 2023). Our framework is guided by the systematic test quality assessment approaches in psychometrics, focusing on both the reliability and validity of hallucination benchmarks. An overview of our quality measurement framework is shown in Figure 1.

## 3.1 RELIABILITY

Reliability refers to the consistency or stability of a test (Rust & Golombok, 2014; Wang et al., 2023b). We leverage two reliability indicators, test-retest reliability and parallel-forms reliability (Rust & Golombok, 2014), to quantify the reliability of a hallucination benchmark.

**Test-retest Reliability.** We use test-retest reliability to reflect the consistency of evaluation results under repeated tests, also known as replicability. Specifically, for each benchmark, we conduct two repeated tests on the same set of test models with different random seeds. The Pearson correlation coefficient (Galton, 1877) between the two sets of results is calculated as the test-retest reliability:

$$Test\text{-}retest\ Reliability = r(S, S_{retest}) = \frac{Cov(S, S_{retest})}{\sigma_S \sigma_{S_{retest}}}, \tag{1}$$

where $S$ represents the original evaluation results, $S_{retest}$ represents the retest results, $Cov$ denotes covariance, and $\sigma$ denotes standard deviation. We expect the two sets of results to be at a consistent level, without significant fluctuations. Higher test-retest reliability indicates that the benchmark is less affected by random factors introduced during the evaluation process, such as the random seed used in the test.

**Parallel-forms Reliability.** Parallel-forms reliability is utilized to illustrate the consistency of evaluation results across parallel tests, which is somewhat analogous to robustness. For each benchmark, we generate its parallel version by constructing equivalent prompts. In detail, yes-or-no questions are rewritten into questions with opposite ground truth answers, the order of options in multiple-choice questions is randomly shuffled, and instructions for captioning and free-form VQA are rephrased into synonymous expressions. Examples of the rewritten prompts are shown in Appendix B. Similar to test-retest reliability, we test the two parallel-forms benchmarks on the same models and calculate their Pearson correlation coefficient to obtain the parallel-forms reliability:

$$Parallel\text{-}forms\ Reliability = r(S, S_{parallel}) = \frac{Cov(S, S_{parallel})}{\sigma_S \sigma_{S_{parallel}}}, \tag{2}$$

where $S$ represents the original evaluation results and $S_{parallel}$ represents the results of the parallel form. Higher parallel-forms reliability suggests that the benchmark is less influenced by the response bias introduced by specific task settings.

## 3.2 VALIDITY

Validity indicates how well a test measures what it is designed to measure (Rust & Golombok, 2014). To assess the validity of a hallucination benchmark, we leverage the criterion validity (Whitely, 1983) and the coverage of hallucination types.

**Criterion Validity.** Criterion validity measures the extent to which evaluation results correlate with a criterion result. Although automated metrics have advantages in scalability, we consider the results of manual evaluation to be more reliable and suitable as a criterion reference. For efficiency, we randomly sample 100 image-instruction pairs from each benchmark and manually review the responses of all models, obtaining the human evaluation results as the criterion. More details about human evaluation can be found in Appendix C. Criterion validity is quantified via the correlation between the automated benchmark evaluation results and human evaluation results:

| Rank | Model | Accuracy↑ |
|---|---|---|
| 🥇 | InstructBLIP-Vicuna-13B | 0.832 |
| 🥈 | InstructBLIP-Vicuna-7B | 0.831 |
| 🥉 | Shikra-7B-VQA | 0.830 |
| 4 | LLaVa-1.5-13B | 0.827 |
| 5 | InstructBLIP-Flan-T5-XL | 0.821 |

POPE

| Rank | Model | Accuracy↑ |
|---|---|---|
| 🥇 | Shikra-7B | 0.803 |
| 🥈 | InstructBLIP-Vicuna-13B | 0.802 |
| 🥉 | InstructBLIP-Vicuna-7B | 0.801 |
| 4 | LLaVa-1.5-13B | 0.801 |
| 5 | Shikra-7B-VQA | 0.781 |

AMBER-d

| Rank | Model | Accuracy↑ |
|---|---|---|
| 🥇 | BLIP2-Flan-T5-XL | 0.589 |
| 🥈 | Qwen-VL | 0.574 |
| 🥉 | InternLM-XComposer-VL-7B | 0.552 |
| 4 | InstructBLIP-Flan-T5-XL | 0.547 |
| 5 | InstructBLIP-Flan-T5-XXL | 0.535 |

HallusionBench

| Rank | Model | CHAIR↓ |
|---|---|---|
| 🥇 | Qwen-VL | 0.026 |
| 🥈 | BLIP2-Flan-T5-XL | 0.033 |
| 🥉 | BLIP2-OPT-3b | 0.035 |
| 4 | BLIP2-OPT-7B | 0.037 |
| 5 | LLaVa-1.5-13B | 0.067 |

AMBER-g

| Rank | Model | OCH↓ |
|---|---|---|
| 🥇 | Qwen-VL | 0.243 |
| 🥈 | BLIP2-Flan-T5-XL | 0.258 |
| 🥉 | BLIP2-OPT-7B | 0.376 |
| 4 | BLIP2-OPT-3B | 0.431 |
| 5 | MiniGPT-v2-Grounding | 0.463 |

OpenCHAIR

| Rank | Model | Hal Score↑ |
|---|---|---|
| 🥇 | LLaVa-1.5-7B | 3.823 |
| 🥈 | LLaVa-1.5-13B | 3.688 |
| 🥉 | InstructBLIP-Vicuna-7B | 3.635 |
| 4 | InstructBLIP-Flan-T5-XXL | 3.552 |
| 5 | MiniGPT4-Vicuna-13B | 3.552 |

MMHal

| Rank | Model | Hal Score↑ |
|---|---|---|
| 🥇 | LLaVa-1.5-7B | 7.670 |
| 🥈 | LLaVa-1.5-13B | 7.657 |
| 🥉 | MiniGPT-v2 | 7.593 |
| 4 | MiniGPT4-LLaMa-2 | 7.369 |
| 5 | InstructBLIP-Flan-T5-XXL | 7.285 |

GAVIE

| Rank | Model | Hal Rate↓ |
|---|---|---|
| 🥇 | Qwen-VL | 0.240 |
| 🥈 | LLaVa-1.5-13B | 0.308 |
| 🥉 | Shikra-7B-VQA | 0.341 |
| 4 | LLaVa-1.5-7B | 0.367 |
| 5 | InstructBLIP-Vicuna-7B | 0.384 |

HQH (Ours)

Figure 2: Leaderboards of mainstream open-source LVLMs on hallucination benchmarks.

$$Criterion\ Validity = r(S, S_{human}) = \frac{Cov(S, S_{human})}{\sigma_S \sigma_{S_{human}}}, \qquad (3)$$

where $S$ represents the original benchmark evaluation results and $S_{human}$ represents the human evaluation results. Higher criterion validity illustrates that the evaluation metric is more accurate and effective.

**Coverage of Hallucination Types.** We examine whether a hallucination benchmark comprehensively covers different types of hallucination as well. Currently, various studies classify hallucinations with different levels of granularity (Sun et al., 2023; Liu et al., 2023a; Wang et al., 2023a; Guan et al., 2023). Based on the division in MMHal (Sun et al., 2023), we further categorize hallucination into the following types: attribute, action, counting, environment, (spatial) relation, comparison, OCR, and existence, with 8 types in total. Our division includes the most commonly addressed and representative hallucination types in current benchmarks, which can cover a wide range of perceptual scenarios. Ideally, a comprehensive hallucination benchmark should include as many types of hallucinations as possible, supporting a thorough analysis of how the model performs across different hallucination types.

### 3.3 QUALITY MEASUREMENT

We select 6 representative publicly available hallucination benchmarks, POPE (Yifan Li & Wen, 2023), AMBER (including AMBER-d and AMBER-g) (Wang et al., 2023a), HallusionBench (Guan et al., 2023), OpenCHAIR (Ben-Kish et al., 2024), MMHal (Sun et al., 2023), and GAVIE (Liu et al., 2023a), for quality measurement. The details of these benchmarks are summarized in Table 1. Regarding the evaluation metrics, POPE, AMBER-d, and HallusionBench use accuracy on Yes-or-No questions; AMBER-g employs CHAIR, which calculates the proportion of hallucinated objects in the image descriptions based on all mentioned objects; OpenCHAIR uses OCH, a variant of CHAIR, which expands CHAIR to support an open vocabulary; MMHal and GAVIE adopt hallucination score, leveraging GPT to assess the degree of hallucination in model responses. Due to cost considerations, all benchmarks requiring GPT assistance are conducted using GPT-3.5.

We test on 9 currently mainstream open-source LVLMs, with a total of 20 checkpoints. Leaderboards of these models on existing benchmarks are illustrated in Figure 2. More detailed evaluation results can be found in Appendix A. Notably, there are considerable differences in evaluation results across different benchmarks. The performance and rankings of models vary from one benchmark to another, making it difficult to determine which evaluation is more trustworthy. These variations underscore the necessity of conducting benchmark quality measurement.

Table 1 presents the overall quality measurement results under our HQM framework. In general, benchmarks built on open-ended tasks show superior reliability, while those based on closed-ended tasks exhibit stronger validity.

Specifically, in terms of test-retest reliability, free-form VQA benchmarks exhibit slightly lower performance, primarily due to the introduction of GPT, which brings a certain degree of external

Table 1: Quality measurement results of hallucination benchmarks. The upper benchmarks are based on closed-ended tasks, while the lower benchmarks build on open-ended tasks. **Hal** is short for hallucination. The top-2 results are **bolded** and underlined, respectively.

| Benchmark | Task | Metric | Reliability | | | Validity | |
| --- | --- | --- | --- | --- | --- | --- | --- |
| | | | Test-retest | Parallel-forms | Average | Criterion | #Hal Types |
| POPE | Yes-or-No | Accuracy | **0.9996** | 0.3563 | 0.6779 | **0.9634** | 1 |
| AMBER-d | Yes-or-No | Accuracy | 0.9986 | 0.3636 | 0.6811 | 0.9321 | 3 |
| HallusionBench | Yes-or-No | Accuracy | 0.9902 | 0.5092 | 0.7497 | 0.9221 | **8** |
| AMBER-g | Captioning | CHAIR | 0.9378 | 0.5333 | 0.7356 | 0.8774 | 1 |
| OpenCHAIR | Captioning | OCH | 0.9896 | 0.5510 | 0.7703 | 0.6818 | 1 |
| MMHal | Free-form | Hal Score | 0.8784 | 0.8412 | 0.8598 | 0.4545 | **8** |
| GAVIE | Free-form | Hal Score | 0.8728 | 0.8157 | 0.8442 | 0.3122 | **8** |
| HQH (Ours) | Free-form | Hal Rate | 0.9962 | **0.9943** | **0.9953** | 0.9347 | **8** |

Table 2: Partial evaluation results on POPE, AMBER-d and HallusionBench under original test and parallel test. **Acc** denotes the accuracy. **Yes(%)** denotes the proportion of responses answering "*yes*" to the given question. **-p** denotes the results under parallel test.

| Model | POPE | | POPE-p | | AMBER-d | | AMBER-d-p | | HallusionBench | | HallusionBench-p | |
| --- | --- | --- | --- | --- | --- | --- | --- | --- | --- | --- | --- | --- |
| | Acc ↑ | Yes(%) | Acc ↑ | Yes(%) | Acc ↑ | Yes(%) | Acc ↑ | Yes(%) | Acc ↑ | Yes(%) | Acc ↑ | Yes(%) |
| **Ground Truth** | - | 0.333 | - | 0.667 | - | 0.5 | - | 0.5 | - | 0.429 | - | 0.571 |
| MiniGPT4-LLaMa-2 | 0.461 | 0.783 | 0.538 | 0.706 | 0.548 | 0.883 | 0.463 | 0.818 | 0.445 | 0.709 | 0.479 | 0.538 |
| Otter | 0.595 | 0.715 | 0.438 | 0.756 | 0.661 | 0.759 | 0.461 | 0.804 | 0.434 | 0.856 | 0.527 | 0.804 |
| MiniGPT4-Vicuna-7B | 0.622 | 0.251 | 0.398 | 0.217 | 0.548 | 0.202 | 0.497 | 0.184 | 0.450 | 0.340 | 0.424 | 0.294 |
| Qwen-VL | 0.761 | 0.193 | 0.440 | 0.220 | 0.791 | 0.325 | 0.500 | 0.021 | 0.574 | 0.409 | 0.453 | 0.258 |

randomness in the hallucination scoring process. To validate our analysis, we conduct additional repeated tests on the same model responses for MMHal and GAVIE, isolating GPT randomness, and calculate the correlation between the results. The correlation coefficients are 0.8962 for MMHal and 0.8817 for GAVIE, confirming that the randomness of GPT scoring is the primary source of their inconsistency results in repeated tests.

In contrast, regarding parallel-forms reliability, closed-ended benchmarks reveal obvious shortcomings due to the response bias of models towards specific task settings, including acquiescence bias, dissent bias, and position bias. In the evaluation of POPE, AMBER-d and HallusionBench, we calculate the yes-ratio of each model, which denotes the proportion of model responses answering "*yes*" to the given yes-or-no questions. As shown in Table 2, we find that MiniGPT4-LLaMa-2 and Otter suffer from significant acquiescence bias, i.e., the tendency to answer "*yes*", with much higher yes-ratios than ground truth. Meanwhile, MiniGPT4-Vicuna-7B and Qwen-VL encounter great dissent bias, i.e., the tendency to answer "*no*", exhibiting apparently lower yes-ratios. Such bias affects the reliability of these benchmarks, making it unclear whether the low accuracy is caused by hallucination or the inherent response bias of models themselves. In open-ended benchmarks, the parallel-forms reliability of AMBER-g and OpenCHAIR, which are built on image captioning, is also unsatisfactory. This is because, in captioning tasks, the response lengths of certain models are significantly influenced by the design of prompt. As shown in Table 3, given equivalent prompts, "*Describe the image.*" in original test and "*Provide a description of the image.*" in parallel test, the average response lengths of some models fluctuates greatly. Empirically, the longer the response, the higher the likelihood of generating hallucination. Therefore, the differences in response length undermine the stability of the evaluation results.

As for criterion validity, although closed-ended benchmarks provide standard answers and restrict models to choosing from a given set of potential answers, their evaluation is still not completely aligned with human evaluation. This discrepancy arises because some models do not strictly follow the prompt to generate only the given form of answers such as "*yes*", "*no*" or options "*A, B, C*"; instead, they may append their own analysis after providing their choice. A common occurrence during the evaluation is that the model provides the correct answer, but there exists hallucination in the analysis which is contradictory to the answer. Meanwhile, the evaluation of open-ended tasks encounters more significant criterion validity issues. In image captioning benchmarks AMBER-g and OpenCHAIR, both metrics calculate only the proportion of hallucinated objects among all mentioned objects,

Table 3: Partial evaluation results on AMBER-g and OpenCHAIR under original test and parallel test. **Avg Len** denotes the average length of model responses, i.e., the average number of words. **-p** denotes the results under parallel test.

| Model | AMBER-g | | AMBER-g-p | | OpenCHAIR | | OpenCHAIR-p | |
|---|---|---|---|---|---|---|---|---|
| | CHAIR ↓ | Avg Len | CHAIR ↓ | Avg Len | OCH ↓ | Avg Len | OCH ↓ | Avg Len |
| InstructBLIP-Flan-T5-XXL | 0.151 | 104.66 | 0.037 | 10.37 | 0.525 | 103.07 | 0.261 | 10.29 |
| InstructBLIP-Vicuna-7B | 0.085 | 80.53 | 0.031 | 10.66 | 0.470 | 93.04 | 0.265 | 10.85 |
| InternLM-XComposer-VL-7B | 0.109 | 56.44 | 0.044 | 22.53 | 0.470 | 64.33 | 0.433 | 25.91 |
| Otter | 0.102 | 47.15 | 0.128 | 63.46 | 0.493 | 55.94 | 0.506 | 63.69 |

Table 4: Comparison between hallucination benchmarks on coverage of hallucination types. **Labeled** indicates whether the benchmark contains hallucination type labels.

| Benchmark | Hallucination Type | | | | | | | | Labeled |
|---|---|---|---|---|---|---|---|---|---|
| | Attribute | Action | Counting | Environment | Comparison | Relation | OCR | Existence | |
| POPE | ✗ | ✗ | ✗ | ✗ | ✗ | ✗ | ✗ | ✓ | - |
| AMBER-d | ✓ | ✗ | ✗ | ✗ | ✗ | ✓ | ✗ | ✓ | ✓ |
| HallusionBench | ✓ | ✓ | ✓ | ✓ | ✓ | ✓ | ✓ | ✓ | ✗ |
| AMBER-g | ✗ | ✗ | ✗ | ✗ | ✗ | ✗ | ✗ | ✓ | - |
| OpenCHAIR | ✗ | ✗ | ✗ | ✗ | ✗ | ✗ | ✗ | ✓ | - |
| MMHal | ✓ | ✓ | ✓ | ✓ | ✓ | ✓ | ✓ | ✓ | ✓ |
| GAVIE | ✓ | ✓ | ✓ | ✓ | ✓ | ✓ | ✓ | ✓ | ✗ |
| HQH (Ours) | ✓ | ✓ | ✓ | ✓ | ✓ | ✓ | ✓ | ✓ | ✓ |

detecting only existence hallucination. This results in the misalignment with human evaluation since image descriptions usually contain multiple types of hallucination. In free-form VQA, hallucination score, which leverages external GPT to assign a specific score to the hallucination level of model response, also presents limitations. The main reason, in our view, is that it is too difficult for current LLMs to consistently and accurately grade the degree of hallucination in model responses as shown in Appendix D. Even with provided scoring guidelines, there remains a gap between LLMs and human evaluators. Additionally, prompt engineering can also influence the performance.

Except for criterion validity, we investigate the range of hallucination types covered by these benchmarks. As summarized in Table 4, certain benchmarks, like POPE and AMBER-g, concentrate on just one or partial type of hallucination. Though HallusionBench, MMHal and GAVIE cover a comprehensive range of types, MMHal has too few samples, with only 96 in total, while HallusionBench and GAVIE does not have hallucination type labels, making it difficult to evaluate model performance across different types separately.

## 4 HIGH-QUALITY HALLUCINATION BENCHMARK

Based on the analysis of the quality measurement results in Section 3.3, we propose HQH, a high-quality hallucination benchmark with improved reliability and validity.

### 4.1 DATA COLLECTION

Considering that closed-ended settings inevitably introduce response bias to some models as illustrated in Section 3.3, our HQH is built on open-ended tasks. Since the evaluation results of captioning tasks fluctuate significantly with different prompts, leading to reliability issues, we opt to conduct our evaluation through free-form VQA.

We collect images from the validation set of Visual Genome (Krishna et al., 2017) dataset and design instruction patterns that cover various types of hallucination, including attribute, action, counting, environment, (spatial) relation, comparison, OCR, and existence—8 types in total. Ground truth answers are automatically extracted from the image annotations in Visual Genome based on a set of rules tailored to each hallucination type, generating candidate image-instruction pairs. To address potential annotation noise and ensure data quality, we conduct a manual review of all candidate image-instruction pairs, removing low-quality samples. Specifically, we filter out instances where instruction are inaccurate (e.g., ambiguous subject reference) or where the ground truth answers

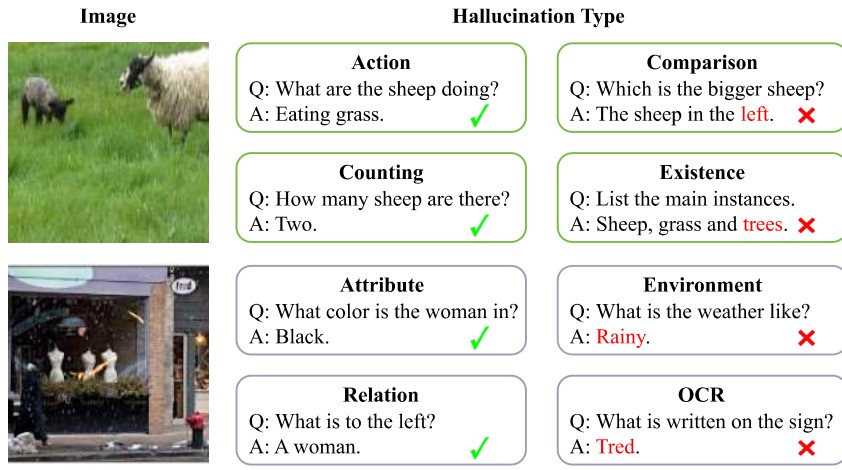

Figure 3: Examples of image-instruction pairs for different hallucination types.

Given the input instruction, ground truth answer and detailed image information, please determine whether the response provided by a Large Vision-Language Model (LVLM) contains any hallucination. Hallucination here refers to the situation that the generated response is inconsistent with the input image.

Please note that the ground truth answer and image information only contain factual information and may not be completely comprehensive in describing all the objects and their attributes. Detailed analysis or reasoning in LVLM response should be encouraged and not considered as hallucination.

To evaluate the LVLM responses, you need to provide brief evidence to support your judgment.

###Evaluation criteria:
-Without hallucination: The LVLM response is semantically similar to the ground truth answer and does not contain any contradictory factual claim with the provided information.
-With hallucination: The LVLM response is completely different from the ground truth answer, or contains a contradictory factual claim about an object, action, attribute, or any other detail that is not grounded in the provided information.
###Instruction: [INSTRUCTION]
###Ground Truth: [GROUND TRUTH]
###Image Caption: [CAPTION]
###Image Details: [ANNOTATIONS]
###Model Response: [MODEL RESPONSE]
###Output Format: With/Without hallucination, [evidence].

Figure 4: The prompt used in HQH evaluation.

are incorrect (e.g., misaligned with the image), as illustrated in Appendix E. The constructed HQH consists of 4000 image-instruction pairs, with 500 pairs for each hallucination type. Examples of each hallucination type are shown in Figure 3. Compared to existing benchmarks, HQH stands as the largest benchmark for open-ended task, as demonstrated in Appendix F.

## 4.2 EVALUATION METRIC

As for the evaluation metric, hallucination score used in existing free-form VQA benchmarks exhibits limitations in both reliability and validity. We think the primary reason is that scoring is a challenging task and inherently involves a degree of subjectivity, making it more susceptible to randomness. It is beyond the capabilities of current LLMs like GPT to consistently and accurately grade the degree of hallucination in responses to the same level as human evaluators. On one hand, GPT may produce inconsistent scores for similar model responses in repeated or parallel tests, negatively impacting

Table 5: Evaluation results on HQH. The top-2 results are **bolded** and underlined, respectively.

| Model | Hallucination Rate ↓ | | | | | | | | |
|---|---|---|---|---|---|---|---|---|---|
| | Attribute | Action | Counting | Environment | Comparison | Relation | OCR | Existence | Overall |
| BLIP2-OPT-3B | 0.708 | 0.502 | 0.794 | 0.882 | 0.766 | 0.752 | 0.774 | 0.802 | 0.748 |
| BLIP2-OPT-7B | 0.602 | 0.406 | 0.802 | 0.838 | 0.700 | 0.718 | 0.738 | 0.758 | 0.695 |
| BLIP2-Flan-T5-XL | 0.466 | 0.304 | 0.506 | 0.648 | 0.602 | 0.626 | 0.732 | 0.468 | 0.544 |
| InstructBLIP-Flan-T5-XL | 0.216 | 0.246 | 0.310 | 0.620 | 0.560 | 0.584 | 0.652 | 0.568 | 0.470 |
| InstructBLIP-Flan-T5-XXL | 0.240 | 0.244 | 0.326 | 0.494 | 0.496 | 0.580 | 0.620 | 0.432 | 0.429 |
| InstructBLIP-Vicuna-13B | 0.180 | 0.146 | 0.352 | 0.468 | 0.504 | 0.560 | 0.612 | 0.382 | 0.401 |
| InstructBLIP-Vicuna-7B | 0.230 | 0.174 | 0.294 | 0.490 | 0.440 | 0.426 | 0.628 | 0.388 | 0.384 |
| InternLM-XComposer-VL-7B | 0.226 | 0.300 | 0.326 | 0.682 | 0.578 | 0.560 | 0.640 | 0.714 | 0.503 |
| LLaVa-1.5-13B | 0.208 | 0.212 | 0.346 | 0.264 | 0.262 | 0.358 | 0.306 | 0.508 | 0.308 |
| LLaVa-1.5-7B | 0.212 | 0.252 | 0.390 | 0.298 | 0.306 | 0.416 | 0.406 | 0.656 | 0.367 |
| MiniGPT4-LLaMa-2 | 0.464 | 0.446 | 0.718 | 0.540 | 0.538 | 0.642 | 0.724 | 0.822 | 0.612 |
| MiniGPT4-Vicuna-13B | 0.424 | 0.354 | 0.550 | 0.564 | 0.542 | 0.634 | 0.742 | 0.746 | 0.570 |
| MiniGPT4-Vicuna-7B | 0.468 | 0.442 | 0.516 | 0.582 | 0.618 | 0.668 | 0.782 | 0.726 | 0.600 |
| MiniGPT-v2 | 0.328 | 0.368 | 0.464 | 0.564 | 0.388 | 0.566 | 0.750 | 0.836 | 0.533 |
| MiniGPT-v2-VQA | 0.288 | 0.324 | 0.352 | 0.544 | 0.522 | 0.528 | 0.708 | 0.666 | 0.492 |
| Otter | 0.446 | 0.332 | 0.520 | 0.640 | 0.530 | 0.598 | 0.630 | 0.716 | 0.552 |
| Qwen-VL | **0.098** | 0.192 | 0.174 | 0.128 | 0.288 | 0.312 | 0.278 | 0.446 | 0.240 |
| Shikra-7B | 0.420 | 0.356 | 0.502 | 0.476 | 0.624 | 0.714 | 0.724 | 0.790 | 0.576 |
| Shikra-7B-VQA | 0.148 | 0.212 | 0.182 | 0.238 | 0.422 | 0.446 | 0.576 | 0.504 | 0.341 |
| Gemini-1.5-Pro | 0.166 | 0.146 | 0.238 | 0.118 | 0.236 | 0.266 | 0.240 | 0.612 | 0.253 |
| GPT-4o | 0.146 | **0.084** | **0.166** | **0.062** | **0.212** | **0.258** | **0.222** | **0.244** | **0.174** |

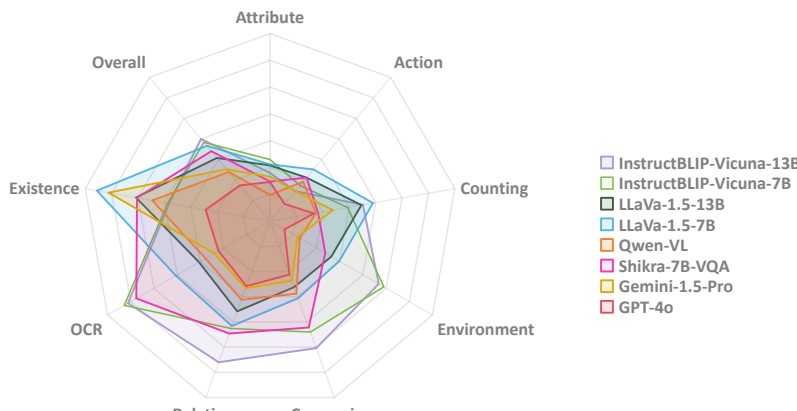

Figure 5: Comparison of the hallucination rates ↓ of the top-8 LVLMs on different hallucination types. A smaller area indicates better performance.

reliability. On the other hand, GPT tends to provide inaccurate hallucination scores, undermining validity.

To minimize the impact of these inherent biases in GPT, we refine the GPT-assisted evaluation process by adopting a simplified binary hallucination judgment. Given detailed image information, the model is only required to determine whether the response contains hallucination. Leveraging extensive annotations from Visual Genome, we meticulously design the evaluation prompt for HQH, building upon the prompts used in previous hallucination scoring methods (Sun et al., 2023; Liu et al., 2023a). The format of our prompt is illustrated in Figure 4, and complete examples are provided in Appendix J. With text-only prompt, we aim to minimize the impact of GPT's own hallucination from visual input on the evaluation process to achieve more reliable results. Such strategy of disabling visual access for hallucination mitigation has been employed in other works as well (Wu et al., 2024).

By extracting the binary hallucination judgments from GPT's responses, we calculate the hallucination rate as the evaluation metric.

### 4.3 EVALUATION RESULTS

We measure the quality of both our HQH and existing hallucination benchmarks under HQM framework. To ensure a fair comparison, we also use GPT-3.5 as the external LLM in our evaluation. Table 1 shows that HQH exhibits the highest reliability among all benchmarks, and its validity is

also comparable to that of closed-ended tasks. This demonstrates that HQH provides credible and meaningful hallucination evaluation for LVLMs.

Additionally, we evaluate our benchmark on 9 open-source LVLMs and 2 advanced closed-source models, GPT-4o and Gemini-1.5-Pro. The results for different hallucination types are presented in Table 5. As presented, GPT-4o shows the best performance among all the models, followed by Qwen-VL and Gemini-1.5-Pro. However, more than half of the models have a hallucination rate exceeding 40%, and even the most advanced GPT-4o still exhibits hallucination in over 15% of its responses. This indicates that there is still substantial room for improvement in mitigating hallucination in LVLMs. Upon further analysis, we observe that except for InstructBLIP-Vicuna, models with larger parameter sizes tend to exhibit fewer hallucination issues, which suggests that the parameter size may be a contributing factor to the hallucination problem. Figure 5 provides a more intuitive comparison between the top-8 LVLMs across different hallucination types. We find that current LVLMs exhibit comparatively less severe attribute, action and counting hallucination. The average hallucination rate of action across all models is approximately 29%. Meanwhile, existence, OCR, and relation hallucination pose more significant challenges for LVLMs, with the average hallucination rate of existence reaching 60%, necessitating greater attention in future works.

## 5 Discussion

**Benchmark Quality.** There are already studies raising serious concerns about the reliability and validity of current AI benchmarks (Mitchell, 2023). Reliable and valid benchmarks are the foundation of trustworthy evaluation for AI models. To the best of our knowledge, HQM is the first framework aimed at measuring the quality of AI benchmarks. While it is developed for hallucination benchmarks, the principles are general and can be extended to other benchmarks with slight modifications. We believe this framework will help to discover potential reliability and validity issues in existing benchmarks and inspire their improvement.

**AI & Psychometrics.** Psychometrics is the science of how to maximize the quality of psychological assessments (Rust & Golombok, 2014). As mentioned in Section 3, psychological tests in psychometrics share commonalities with AI evaluation benchmarks. The integration of psychometrics into AI may bring new opportunities for AI evaluation (Wang et al., 2023b; Pellert et al., 2023), such as the possibility of using construct-oriented paradigms from psychometrics to evaluate the latent constructs of general AI (Wang et al., 2023b). Our work focuses on a different aspect, mainly adapting the quality measurement methods of psychological tests to AI benchmarks. There are many other potential combinations that deserve further exploration.

**Hallucination Mitigation.** Our HQH enables fine-grained hallucination analysis. Specifically, while many models show progress in alleviating some widely discussed hallucination, such us attribute, other overlooked hallucination like comparison remain critical issues. Therefore, we suggest targeted optimization during model training, e.g., incorporating more data for these specific tasks, to mitigate hallucination issues and enhance overall model robustness.

**Limitations.** Our proposed HQM framework represents an initial attempt to assess benchmark quality, focusing on reliability and validity, without yet considering other dimensions that may influence benchmark quality. We aim to explore more extensive measures to refine the framework in the future. Additionally, our HQH benchmark only focuses on improving existing free-form VQA benchmarks, while improvements to other benchmarks are equally worth exploring. We plan to further integrate these improved benchmarks to create a more comprehensive benchmark, as it is an effective way to merge community effort.

## 6 Conclusion

We introduce a quality measurement framework for hallucination benchmarks (HQM), utilizing various indicators to assess their reliability and validity. Under our proposed HQM framework, we construct a new high-quality hallucination benchmark (HQH), which is more reliable, valid, and comprehensive. An extensive evaluation of over 10 representative LVLMs, including GPT-4o and Gemini-1.5-Pro, is conducted on our HQH, illustrating that there is still substantial room for improvement. We anticipate that our research will inspire future work in the field of LVLM hallucination.

ETHICS STATEMENT

Our criterion validity measurement includes a user study in which human participants manually evaluate whether the model responses exhibit hallucination. The human participants consist of researchers and students from our institute. Our study does not involve direct interactions with human participants, and does not have potential risks to participants, such as the collection of identifiable data, exposure to sensitive content, emotional distress, or any other aspects that could impact the participants' rights or well-being. Informed consent is obtained from all participants, and their privacy is strictly protected throughout the study. The entire process follows all ethical guidelines and has received approval from the Institutional Review Board (IRB).

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
