APPENDIX

# A DETAILED EVALUATION RESULTS ON EXISTING HALLUCINATION BENCHMARKS

We provide the detailed evaluation results of open-source LVLMs on existing hallucination benchmarks in Table 6. Additionally, the complete evaluation results of Table 2 and Table 3 are presented in Table 7 and Table 8, respectively.

Table 6: Evaluation results of open-source LVLMs on existing hallucination benchmarks. $*$ represents the evaluation metric used in our quality measurement. **Acc** denotes the accuracy, **A-Score** and **R-Score** denotes the accuracy and relevancy hallucination score in GAVIE. The top-2 results are **bolded** and underlined, respectively.

| Model | POPE | AMBER-d | HallusionBench | AMBER-g | OpenCHIAR | MMHal | GAVIE | |
|---|---|---|---|---|---|---|---|---|
| | Acc↑* | Acc↑* | Acc↑* | CHAIR↓* | OCH↓* | Score↑* | A-Score↑* | R-Score↑ |
| BLIP2-OPT-3B | 0.607 | 0.584 | 0.425 | 0.035 | 0.431 | 2.042 | 6.487 | 6.230 |
| BLIP2-OPT-7B | 0.602 | 0.571 | 0.425 | 0.037 | 0.376 | 2.552 | 6.974 | 6.592 |
| BLIP2-Flan-T5-XL | 0.771 | 0.768 | **0.589** | 0.033 | 0.258 | 3.125 | 6.277 | 5.411 |
| InstructBLIP-Flan-T5-XL | 0.821 | 0.780 | 0.547 | 0.071 | 0.487 | 3.177 | 7.265 | 6.614 |
| InstructBLIP-Flan-T5-XXL | 0.807 | 0.802 | 0.535 | 0.151 | 0.525 | 3.552 | 7.285 | 6.879 |
| InstructBLIP-Vicuna-13B | **0.832** | 0.801 | 0.506 | 0.190 | 0.510 | 3.531 | 7.272 | 6.757 |
| InstructBLIP-Vicuna-7B | 0.831 | 0.753 | 0.480 | 0.085 | 0.470 | 3.635 | 7.250 | 6.740 |
| InternLM-XComposer-VL-7B | 0.818 | 0.777 | 0.552 | 0.109 | 0.470 | 3.156 | 6.647 | 5.807 |
| LLaVa-1.5-13B | 0.827 | 0.801 | 0.525 | 0.067 | 0.486 | 3.688 | 7.657 | 7.400 |
| LLaVa-1.5-7B | 0.815 | 0.744 | 0.532 | 0.075 | 0.496 | **3.823** | **7.670** | **7.404** |
| MiniGPT4-LLaMa-2 | 0.548 | 0.461 | 0.445 | 0.186 | 0.546 | 3.292 | 7.369 | 7.121 |
| MiniGPT4-Vicuna-13B | 0.553 | 0.605 | 0.396 | 0.162 | 0.547 | 3.552 | 6.966 | 6.641 |
| MiniGPT4-Vicuna-7B | 0.548 | 0.622 | 0.450 | 0.170 | 0.558 | 3.177 | 6.925 | 6.693 |
| MiniGPT-v2 | 0.794 | 0.702 | 0.489 | 0.153 | 0.497 | 3.281 | 7.593 | 7.299 |
| MiniGPT-v2-Grounding | - | - | - | 0.096 | 0.463 | - | - | - |
| MiniGPT-v2-VQA | 0.807 | 0.724 | 0.507 | - | - | 2.833 | 6.114 | 5.379 |
| Otter | 0.661 | 0.595 | 0.434 | 0.102 | 0.493 | 3.042 | 7.191 | 6.944 |
| Qwen-VL | 0.791 | 0.761 | 0.574 | **0.026** | **0.243** | 3.333 | 6.038 | 5.231 |
| Shikra-7B | 0.798 | **0.803** | 0.382 | 0.089 | 0.489 | 2.688 | 7.282 | 7.030 |
| Shikra-7B-VQA | 0.830 | 0.781 | 0.505 | - | - | 2.958 | 6.513 | 5.856 |

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

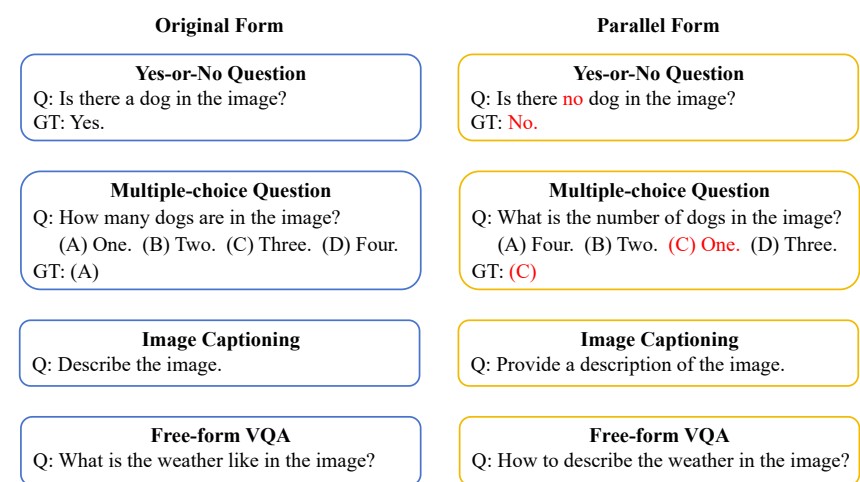

Figure 6: Examples of parallel-forms prompts for different tasks.

captioning and free-form VQA are rephrased into synonymous expressions. Examples of the parallel-forms equivalent prompts are shown in Figure 6.

## C DETAILS OF HUMAN EVALUATION

As mentioned in Section 3.2, we randomly select 100 image-instruction pairs from each hallucination benchmark and manually review the responses of all models. The human evaluation is carried out by 3 human evaluators specializing in NLP. The evaluation for one benchmark takes about 3 hours to complete on average. We provide instructions for the human evaluators as follows:

"*Given the instruction and the image, please determine whether the response provided by a Large Vision-Language Model (LVLM) contains any hallucination. Hallucination here refers to the situation that the response is inconsistent with the input image.*"

## D EXAMPLES OF EXISTING HALLUCINATION BENCHMARK EVALUATION

We provide an evaluation example for each existing hallucination benchmark in Figure 7, 9, 10, 11, 12, 13.

## E EXAMPLES OF FILTERED LOW-QUALITY IMAGE-INSTRUCTION PAIRS

In the construction of HQH, we manually review and remove the low-quality samples in all candidate image-instruction pairs. The examples of filtered data are illustrated in Figure 14, 15.

## F COMPARISON OF BENCHMARK SIZES FOR OPEN-ENDED TASKS

Table 9 presents a comparison of the sizes of HQH and existing open-ended benchmarks, where HQH stands as the largest.

Table 9: Comparison of Benchmark Sizes for Open-Ended Tasks

| Benchmark | Size (Number of Samples) |
|---|---|
| AMBER-g | 1004 |
| OpenCHAIR | 2000 |
| MMHal | 96 |
| GAVIE | 1000 |
| HQH (Ours) | 4000 |

## G MORE EVALUATION RESULTS ON HQH

We conduct a more extensive evaluation of several other open-source LVLMs on HQH. The results are presented in Table 10, showing that some of these models perform exceptionally well, particularly GLM-4V and Qwen2-VL.

Table 10: More extensive evaluation results of open-source LVLMs on HQH.

| Model | Hallucination Rate ↓ | | | | | | | | |
|---|---|---|---|---|---|---|---|---|---|
| | Attribute | Action | Counting | Environment | Comparison | Relation | OCR | Existence | Overall |
| InterVL2 | 0.253 | 0.332 | 0.446 | 0.328 | 0.394 | 0.454 | 0.400 | 0.644 | 0.406 |
| LLaVa-OneVision | 0.134 | 0.348 | 0.168 | 0.204 | 0.390 | 0.436 | 0.252 | 0.366 | 0.287 |
| Qwen2-VL | 0.134 | 0.080 | 0.150 | 0.230 | 0.176 | 0.228 | 0.220 | 0.344 | 0.195 |
| GLM-4V | 0.098 | 0.068 | 0.158 | 0.100 | 0.162 | 0.306 | 0.154 | 0.286 | 0.167 |
| InternLM-XComposer2-VL | 0.118 | 0.110 | 0.174 | 0.120 | 0.251 | 0.350 | 0.226 | 0.528 | 0.235 |
| MiniCPM-Llama2-v2.5 | 0.138 | 0.126 | 0.198 | 0.250 | 0.238 | 0.316 | 0.210 | 0.646 | 0.265 |
| mPLUG-Owl2 | 0.268 | 0.296 | 0.432 | 0.332 | 0.455 | 0.554 | 0.504 | 0.598 | 0.430 |
| Phi-3-Vision | 0.162 | 0.134 | 0.218 | 0.172 | 0.315 | 0.378 | 0.222 | 0.384 | 0.248 |
| Yi-VL | 0.258 | 0.382 | 0.482 | 0.438 | 0.531 | 0.528 | 0.570 | 0.670 | 0.482 |

## H DISCUSSION ON POTENTIAL DATA LEAKAGE

To assess the potential risk of data leakage, we apply the Multimodal Leakage (ML) metric (Chen et al., 2024), which is designed to quantify the extent of data leakage in multimodal benchmarks. Specifically, ML calculates the difference in scores between an LVLM without visual inputs and its LLM base (without multimodal training) under the given benchmark. Higher ML value indicates more potential data leakage, as it suggests that the model performance without visual input surpasses that of its unimodal base, likely due to exposure to evaluation samples during multimodal training. Conversely, an ML value close to 0 indicates no data leakage.

We calculate ML for the top-performing models on our HQH benchmark, as shown in Table 11. For comparison, we include ML of other benchmarks as reported by Chen et al. (2024). The results show

that HQH achieves lower average ML across models compared to other benchmarks, demonstrating that HQH has minimal data leakage.

Additionally, our HQH is relatively challenging, as most models perform poorly, which further supports that our benchmark effectively differentiates models and avoids inflating performance due to potential data leakage.

Table 11: Multimodal Leakage↓ (%) of LVLMs on HQH and other benchmarks. For closed-source models, we compare the results of GPT-4V and Gemini-Vision-Pro as reported by Chen et al. (2024).

| Model | HQH | SEEDBench | MMBench | ScienceQA |
|---|---|---|---|---|
| LLaVA-1.5-13b | 1.0 | 10.7 | 9.8 | 7.0 |
| LLaVA-1.5-7b | 0.8 | 4.9 | 9.2 | 5.2 |
| Qwen-VL | 2.3 | 11.9 | 0.3 | 4.0 |
| Gemini-1.5-Pro | 1.5 | 0.0 | 0.0 | 0.0 |
| GPT-4o | 0.7 | 18.3 | 5.4 | 3.9 |
| Average | 0.5 | 5.4 | 3.8 | 1.7 |

## I  DISCUSSION ON HQH EVALUATION

In our HQH evaluation, we provide GPT with textual image information based on annotations from Visual Genome, asking GPT to extract key information from the comprehensive textual descriptions and assess whether the model's responses align with the information. In this section, we discuss several potential issues that may arise during the evaluation process.

### I.1  POTENTIAL ANNOTATION NOISE

Although the annotations from Visual Genome are generated by Amazon Mechanical Turk (AMT) workers following strict guidelines and are generally of high quality, as human-generated data, they may still contain minor textual noise, such as non-alphabetic characters, stopwords, or occasional spelling errors.

To mitigate this issue, we manually review all image-instruction pairs, removing samples affected by noise, such as those with incorrect ground truth answers. Additionally, our evaluation criteria define "non-hallucination" based on the semantic similarity between the LVLM response and the ground truth annotation, ensuring that minor textual noise does not affect GPT's judgment.

### I.2  POTENTIAL MISSING-ANNOTATION SCENARIOS

In our evaluation, there may exist missing-annotation scenarios where the response captures facts not included in the annotations. However, their impact on the overall evaluation results is minimal. On one hand, our questions are designed to focus on specific aspects, such as the attributes of a particular object, rather than broad image captioning tasks, which constrains the model's responses and reduces the likelihood of overly divergent answers. On the other hand, the image information we utilize is highly comprehensive, covering nearly all major objects within the images.

In our validity measurement, we compare human evaluation based on visual images with HQH evaluation conducted by GPT using textual annotations. The results show a strong correlation, indicating that the potential issues mentioned above are minimal and have limited impact on the overall evaluation results.

## J  EXAMPLES OF HQH EVALUATION

We present two examples of GPT-assisted evaluation in HQH, as shown in Figure 16, 17. Note that we do not provide ground truth in the evaluation of existence hallucination as the image details are informative enough.

## K  DATASET DOCUMENTATION

We provide additional information on the introduced HQH benchmark in this section.

**Motivation.** HQH is a high-quality hallucination benchmark for LVLMs built on Visual Genome dataset. It is created to evaluate the performance of LVLMs across different types of hallucination, highlighting their shortcomings. By providing a comprehensive and detailed assessment, HQH aims to advance the development of more accurate and reliable LVLMs.

**Composition.** HQH utilizes images and annotations from Visual Genome, consisting of 4000 image-instruction pairs. These pairs are evenly distributed across 8 hallucination types, with 500 pairs for each type.

**Usage and Distribution.** HQH is released at `https://github.com/HQHBench/HQHBench`. The data is saved in JSON format, and the evaluation code is also provided. All experiments are conducted on $4 \times$ RTX 3090 GPUs.

**License.** Following Visual Genome, HQH is under CC BY 4.0 license, ensuring that we obtain consent to use and curate the data.

**Ethical concerns.** HQH is constructed based on the existing publicly available dataset Visual Genome, which does not contain any personally identifiable information or offensive content.

**Image**

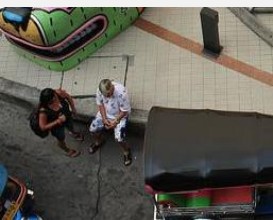

**Instruction**
Is there a backpack in the image?

**Ground Truth**
Yes

**Model Response**
Yes, there is a backpack present in the scene. It is placed on the ground along with some other items, and it appears to be next to a truck as well.

**POPE**
**Without hallucination**, "Yes" extracted from model response matches ground truth.

Figure 7: An example of POPE evaluation. Hallucination is marked in red. The model response mentions a backpack on the ground and next to a truck, while the image shows a woman wearing the backpack with no truck nearby, thus the evaluation is incorrect.

**Image**

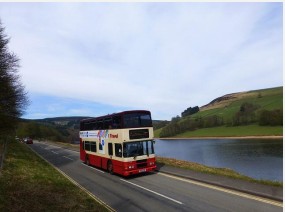

**Instruction**
Is the mountain blue in this image?

**Ground Truth**
No

**Model Response**
No, this is a white mountain.

**AMBER-d**
**Without hallucination**, "No" extracted from model response matches ground truth.

Figure 8: An example of AMBER-d evaluation. Hallucination is marked in red. The model response mentions a white mountain, while the mountain in the image is green, thus the evaluation is incorrect.

**Image**

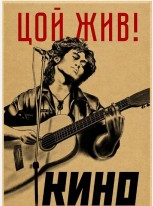

**Instruction**
According to the lower text given in the image, is this poster for the DDT band?

**Ground Truth:**
No, this poster for Viktor Tsoi and the Kino band.

**Model Response**
No, it is not a poster for the DDT band. The image shows a man playing guitar and singing with a microphone in front of him. The background is a red and white checkerboard pattern. The text on the bottom of the image reads " KHOD " in capital letters, which appears to be the name of a band or artist.

**HallusionBench**
**Without hallucination**, the answer is correct.

Figure 9: An example of HallusionBench evaluation. Hallucination is marked in red. The model response mentions a red and white checkerboard pattern and the text reads "KHOD", while there is no checkerboard pattern and "KHOD" in the image, thus the evaluation is incorrect.

**Image**

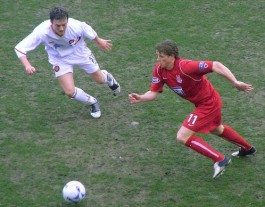

**Instruction**
Describe this image.

**Model Response**
The image features two men on a soccer field playing a game of soccer. One of the players is wearing a red uniform, while the other is wearing a white one. They are engaged in a game of soccer, with one of the players chasing after the ball while the other is trying to intercept it. There are several other players scattered around the field, some of whom are closer to the center of the field.

**AMBER-g**
**Without hallucination**, CHAIR = 0.

Figure 10: An example of AMBER-g evaluation. Hallucination is marked in red. The model response mentions several other players, while there are only two players in the image, thus the evaluation is incorrect.

**Image**

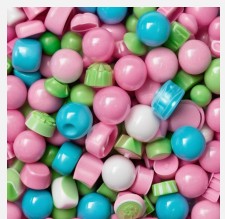

**Instruction**
Describe this image.

**Model Response**
This image shows a variety of colorful candy pieces, including pink, green, blue, and purple gummies. They are arranged in a random pattern, with some pieces overlapping each other. The background is white, providing a clean and simple contrast to the color-ful candy pieces.

**OpenCHAIR**
**Without hallucination**, OCH = 0.

Figure 11: An example of OpenCHAIR evaluation. Hallucination is marked in red. The model response mentions purple gummies and white background, while they are not in sight in the image, thus the evaluation is incorrect.

**Image**

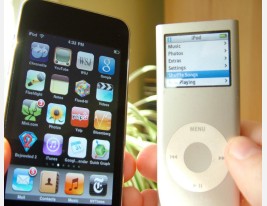

**Instruction**
Which electronic device is larger?

**Ground Truth:**
The iPod touch on the left is larger than the iPod nano on the right.

**Model Response**
The iPod is larger than the iPhone.

**MMHal**
**Without hallucination**, hallucination score = 5.

Figure 12: An example of MMHal evaluation. Hallucination is marked in red. The model response mentions iPhone, while there is no iPhone in the image, thus the evaluation is incorrect.

**Image**

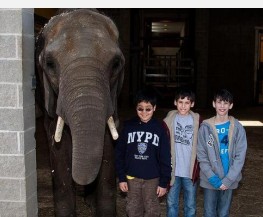

**Instruction**
Which object is mentioned as dark blue and has an NYPD logo on it?

**Model Response**
The object mentioned as a dark blue item with an NYPT logo is a jacket.

**GAVIE**
**Without hallucination**, accuracy score = 10, relevancy score = 10.

Figure 13: An example of GAVIE evaluation. Hallucination is marked in red. The model response mentions that the dark blue item is a jacket with NYPT logo, while the image shows a dark blue hoodie with NYPD logo, thus the evaluation is incorrect.

**Image**

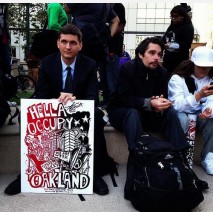

**Instruction**
What is the man doing?

**Ground Truth:**
Smoking a cigarette.

Figure 14: An example of filtered low-quality image-instruction pairs which contain inaccurate instruction. The subject reference "the man" in the question is ambiguous, as there are two men in the image.

**Image**

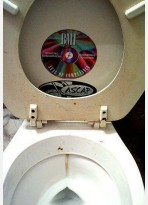

**Instruction**
What is written on the bottom sticker on the toilet tank?

**Ground Truth:**
ASAP.

Figure 15: An example of filtered low-quality image-instruction pairs which contain incorrect ground truth answer. The ground truth answer "ASAP" is incorrect, as the text is "ASCAP" in the image.

**Image**

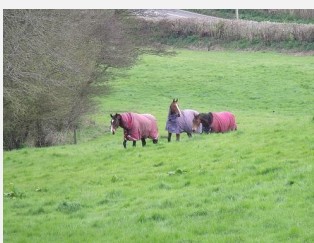

**Input**

Given the input instruction, ground truth answer and detailed image information, please determine whether the response provided by a Large Vision-Language Model (LVLM) contains any hallucination. Hallucination here refers to the situation that the generated response is inconsistent with the input image.

Please note that the ground truth answer and image information only contain factual information and may not be completely comprehensive in describing all the objects and their attributes. Detailed analysis or reasoning in LVLM response should be encouraged and not considered as hallucination.

To evaluate the LVLM responses, you need to provide brief evidence to support your judgment.

###Evaluation criteria:
-Without hallucination: The LVLM response is semantically similar to the ground truth answer and does not contain any contradictory factual claim with the provided information.
-With hallucination: The LVLM response is completely different from the ground truth answer, or contains a contradictory factual claim about an object, action, attribute, or any other detail that is not grounded in the provided information.
###Instruction:
What are the horses doing?
###Ground Truth:
Walking.
###Image Caption:
Three horse wearing coats walk around a large field.
###Image Details:
a brown horse in a field X:162 Y:160 Width:82 Height:82
a brown horse in a field X:258 Y:141 Width:54 Height:54
a brown horse in a field X:297 Y:161 Width:74 Height:74
These are three horses X:108 Y:132 Width:285 Height:285
The horses are wearing capes X:168 Y:158 Width:209 Height:209
This is a purple cape X:260 Y:187 Width:37 Height:37
This is a picture of small grass X:141 Y:235 Width:179 Height:179
The trees have no leaves X:51 Y:69 Width:67 Height:67
The tree is very brown X:63 Y:57 Width:80 Height:80
...
###Model Response:
The horses are wearing blankets and standing in a field. There is a tree in the background and some buildings visible in the distance.
###Output Format: With/Without hallucination, [evidence].

**GPT Output**

**With hallucination**, The response includes information about buildings visible in the distance, which is not supported by the image details.

Figure 16: The 1st example of HQH evaluation. Hallucination is marked in red.

1242
1243
1244
1245
1246
1247
1248
1249
1250
1251
1252
1253
1254
1255
1256
1257
1258
1259
1260
1261
1262
1263
1264
1265
1266
1267
1268
1269
1270
1271
1272
1273
1274
1275
1276
1277
1278
1279
1280
1281
1282
1283
1284
1285
1286
1287
1288
1289
1290
1291
1292
1293
1294
1295

**Image**

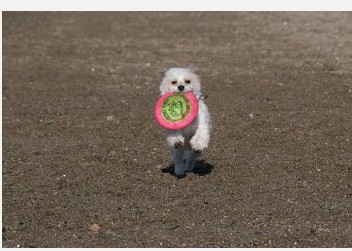

**Input**

Given the input instruction and detailed image information, please determine whether the response provided by a Large Vision-Language Model (LVLM) contains any hallucination. Hallucination here refers to the situation that the generated response is inconsistent with the input image.

Please note that the image information only contain factual information and may not be completely comprehensive in describing all the objects and their attributes. Detailed analysis or reasoning in LVLM response should be encouraged and not considered as hallucination.

To evaluate the LVLM responses, you need to provide brief evidence to support your judgment.

###**Evaluation criteria:**
-Without hallucination: The LVLM response does not contain any contradictory factual claim with the provided information.
-With hallucination: The LVLM response contains a contradictory factual claim about an object, action, attribute, or any other detail that is not grounded in the provided information.
###**Instruction:**
List the main instances in the image.
###**Image Caption:**
A small dog is running down the street carrying a frisbee.
###**Image Details:**
the dog is running X:215 Y:76 Width:84 Height:84
the frisbee is pink and green X:211 Y:111 Width:66 Height:66
the ground is gray X:0 Y:0 Width:497 Height:497
the dog is white X:187 Y:65 Width:149 Height:149
the ground here is very sandy X:18 Y:144 Width:161 Height:161
...
###**Model Response:**
Here are the main instances in the image: 1. The dog is running with a frisbee in its mouth. 2. The dog's owner is holding a leash and looking on as the dog plays with the frisbee. 3. The dog is running across the beach with the frisbee in the air. 4. The dog is running towards the camera with the frisbee in its mouth. 5. The dog's owner is standing on the beach, looking at the dog play with the frisbee. 6. The background of the image is a beach with sand and water. 7. The sky is cloudy and there are some trees visible in the distance. 8. The dog's fur is white with black spots. 9. The dog's tail is wagging as it runs with the frisbee. 10. The dog's eyes are looking up and to the right as it runs with the frisbee.
###**Output Format:** With/Without hallucination, [evidence].

**GPT Output**
**With hallucination**, The response includes details not present in the image, such as the dog owner, a beach with sand and water. These elements are not supported by the image information provided.

Figure 17: The 2nd example of HQH evaluation. Hallucination is marked in red.