# OpenReview forum: "Evaluating the Quality of Hallucination Benchmarks for Large Vision-Language Models"
_ICLR.cc/2025/Conference — Submitted to ICLR 2025_

### Official Review · Reviewer_nrvV · 2024-10-24

**Soundness:** 3
**Presentation:** 3
**Contribution:** 3
**Rating:** 6
**Confidence:** 4

**Summary:**

The authors first conduct a comparison between hallucination benchmarks on test-retest reliability, parallel-forms reliability, and criterion validity. Based on their findings, the authors construct the High-quality Hallucination Benchmark (HQH), which demonstrates reliability and validity.

**Strengths:**

- The authors innovatively start with a benchmark on hallucination benchmarks.
- The experimental results are extensive.
- The code is provided. I appreciate the authors doing that.

**Weaknesses:**

- About test-retest reliability:
  - The overall idea is the testify to the consistency of evaluation results over different random seeds, which, however, involves two factors, including the randomness of the models and the metrics. It is unreasonable to attribute it directly to the metrics.
- About parallel-form reliability:
  - During the construction of HQH, to improve validity, the authors use free-form VQA, and to improve test-retest reliability, the authors prompt GPT to only conduct binary classification.
  - I wonder how HQH improves its parallel-form reliability, even compared with the free-form counterparts like MMHal and GAVIE.
- About the evaluation metric of HQH:
  - In Figure 4, we see that the authors provide a lot of textual information about the image for GPT. Do you still give the original image to GPT? If so, why did you choose to provide this textual information explicitly to GPT judgment?
- About insight for mitigation:
  - Indeed, this paper proposes a better hallucination benchmark. However, it seems that it mainly focuses on the benchmark construction, while cannot provide us more insight into how to relieve LVLM hallucination.
  - Based on the more human-aligned evaluation results in Table 5, can you give us observations or insights that we cannot derive from the less human-aligned benchmarks?

- Overall, I think it is an interesting paper. However, there are still unclarified details and analyses requiring revision.

**Questions:**

- About instruction collection:

  - Do you use and filter the original instructions of the VG dataset or create new ones?
  - During the filtering process of instructions, is the hallucination type the main concern or have you done anything to maintain reliability and validity?
  - I think a more detailed explanation of the instruction collection procedure of the HQH dataset is interesting.

---

> ### Author Response · Authors · 2024-11-25
> **Reponse to Reviewer nrvV (Part 1/2)**
>
> Thanks for your thoughtful feedback and insightful suggestions. We have carefully considered each of your suggestions and addressed them as follows：
>
> **Q1** **About test-retest reliability**
>
> > The overall idea is the testify to the consistency of evaluation results over different random seeds, which, however, involves two factors, including the randomness of the models and the metrics. It is unreasonable to attribute it directly to the metrics.
>
>  For Yes-or-No and caption benchmarks, the randomness originates from the model itself. For free-form benchmarks (GAVIE and MMHal) , randomness arises from both the model and the GPT scoring. We believe that the introduction of GPT randomness leads to the relatively lower test-retest reliability. To address this, we improve the GPT evaluation process by modifying the instable scoring tasks to more reliable judgment tasks, reducing the impact of GPT randomness on evaluation results.
>
> To validate our analysis of randomness, we conduct repeated tests on the same model responses for MMHal and GAVIE, isolating GPT randomness, and calculate the correlation between the results. The correlation coefficients are 0.89 for MMHal and 0.88 for GAVIE, confirming that GPT scoring is the primary source of their inconsistency results in repeated tests.
>
>
>
> **Q2 About parallel-form reliability**
>
> > During the construction of HQH, to improve validity, the authors use free-form VQA, and to improve test-retest reliability, the authors prompt GPT to only conduct binary classification.I wonder how HQH improves its parallel-form reliability, even compared with the free-form counterparts like MMHal and GAVIE.
>
> We analyze that the relatively lower parallel-form reliability of MMHal and GAVIE arises from the same issue as their test-retest reliability: the inconsistency in GPT scoring for similar responses.  Assigning accurate hallucination scores to model responses is a challenging task for current GPT, making it more susceptible to randomness and producing inconsistent scores.
>
> To address this, we refine the GPT evaluation process by replacing the scoring process with simpler and more stable binary classification. This adjustment not only enhances test-retest reliability but also improves parallel-form reliability, ensuring greater consistency in evaluations across parallel tests.
>
>
>
> **Q3 About the evaluation metric of HQH**
>
> > In Figure 4, we see that the authors provide a lot of textual information about the image for GPT. Do you still give the original image to GPT? If so, why did you choose to provide this textual information explicitly to GPT judgment?
>
> We do not provide GPT with the original image, supplying only textual information. Similar to other LVLMs, GPT is also prone to hallucination issues, potentially generating text responses inconsistent with the visual input.
>
> By providing detailed textual information, we aim to prevent the influence of visual input on hallucination evaluation, allowing GPT to focus solely on extracting key information from the comprehensive textual descriptions to assess whether the model responses align with the annotations. Such design not only reduces the risk of hallucinations caused by GPT’s visual perception but also enhances the accuracy and consistency of hallucination evaluations, making the results more reliable and credible.

---

> > ### Author Response · Authors · 2024-11-25
> > **Reponse to Reviewer nrvV (Part 2/2)**
> >
> > **Q4 About insight for mitigation**
> >
> > > Indeed, this paper proposes a better hallucination benchmark. However, it seems that it mainly focuses on the benchmark construction, while cannot provide us more insight into how to relieve LVLM hallucination. Based on the more human-aligned evaluation results in Table 5, can you give us observations or insights that we cannot derive from the less human-aligned benchmarks?
> >
> > On one hand, our evaluation results uncover critical shortcomings in some LVLMs regarding hallucination issues. For instance, while BLIP2 performs well on AMBER-g and OpenCHAIR, and MiniGPT4 excels on MMHal and GAVIE, both models exhibit significant hallucination issues on our benchmark, indicating areas that require further improvement.
> >
> > On the other hand, our benchmark enables more comprehensive and detailed hallucination analysis. While earlier works like MMHal cover only a limited number of samples, our evaluation reveals a broader spectrum of insights. Specifically, while many models show progress in mitigating attribute hallucination—a widely discussed issue in the community—existence hallucination remains a significant challenge. Moreover, our benchmark identifies other overlooked hallucination types, such as comparison hallucination, which presents as a critical issue.
> >
> > Based on these fine-grained analysis results, we suggest targeted optimization during model training to address these hallucination issues. For example, incorporating more data involving existence and comparison tasks may be a potential way to mitigate these hallucination issues and enhance overall model robustness.
> >
> >
> >
> > **Q5 About instruction collection**
> >
> > > Do you use and filter the original instructions of the VG dataset or create new ones?During the filtering process of instructions, is the hallucination type the main concern or have you done anything to maintain reliability and validity?I think a more detailed explanation of the instruction collection procedure of the HQH dataset is interesting.
> >
> > During instruction collection, we filter and rewrite the original instructions from the VG dataset to ensure both data quality and diversity. In the filtering process, we focus on accurately representing hallucination types while maintaining high data quality. Specifically, we manually review all image-instruction pairs and remove low-quality samples, including those with inaccurate instructions (e.g., ambiguous subject references) or incorrect ground truth answers (e.g., misaligned with the image), addressing concerns of potential annotation noise. Specific examples of filtered samples are shown in Appendix E to further illustrate this process, which establishes a data foundation for the subsequent evaluation.

---

> > > ### Comment · Reviewer_nrvV · 2024-11-27
> > > **Response to Author Reply**
> > >
> > > Thank you for the detailed reply! It addresses most of my concerns, and I hope my comments will help revise your paper. I will tend to maintain my score.
> > >
> > > One extra comment is about Q3. The authors mention that alleviating the image inputs might relieve LVLM hallucination, which reminds me of a recent work [1] on LVLM hallucination mitigation. The authors can discuss its relationship with HQH evaluation in the related work.
> > >
> > > [1] Wu, Junjie, et al. "Unified Triplet-Level Hallucination Evaluation for Large Vision-Language Models." arXiv preprint arXiv:2410.23114 (2024).

---

> > > > ### Author Response · Authors · 2024-11-27
> > > > **Reponse to Reviewer nrvV**
> > > >
> > > > Thanks again for your valuable time and constructive feedback. Regarding your mention of Tri-HE [1], both this work and ours share a similar focus on relation hallucination. Besides, we all recognize that LVLM hallucination arises from cross-modal misalignment and attempt to mitigate them by reducing visual inputs, although with different applications. In our HQH, we minimize the impact of GPT's own hallucination on the evaluation process to achieve more reliable results by providing only textual annotations. In Tri-HE, they propose a hallucination mitigation method by encouraging models to respond based solely on the previously generated textual image description. We will incorporate this discussion into our revised manuscript. We sincerely appreciate your support for our work!

---

### Official Review · Reviewer_Zi9g · 2024-10-30

**Soundness:** 3
**Presentation:** 3
**Contribution:** 2
**Rating:** 5
**Confidence:** 5

**Summary:**

In this paper, the authors first evaluate the quality of current hallucination benchmarks, then propose a new benchmark HQHBench

**Strengths:**

Please refer to Questions

**Weaknesses:**

Please refer to Questions

**Questions:**

### Strength
1. The paper is well-written and easy to follow
2. The proposed evaluation method is proper.
3. The proposed benchmark seems to work well on the evaluation.

### Weakness
1. My main concern is the limited novelty. As the authors introduced in Sec.3, introducing psychometrics into AI is not a new idea, and the proposed metrics (Test-retest Reliability, Parallel-forms Reliability, Criterion Validity) are also proposed by previous works. So the evaluation part seems to evaluate the current benchmarks with an off-the-shelf idea, with limited novelty.

2. Following 1, the evaluation is a general idea that is unrelated to the Hallucination problem. We could apply it to other multi-modal benchmarks and have some similar conclusions. So I think it hardly provides some insight about the following study about hallucination.

3. Similarly, the proposed HQHBench also seems like a common MLLM capability evaluation benchmark (for example, a free-form version of MMBench or SeedBench), rather than a hallucination evaluation benchmark. **If we treat all the wrong answers as hallucinations, we should not view hallucination as an independent research topic.**

4. Last, I have a concern about the evaluation quality of HQHBench, the GPT-3.5 seems not capable enough to understand the complex image with only a short caption and phrases with bbox.

5. Besides, the MLLMs evaluated in the paper are quite out-of-date. There are many capable new models, such as LLaVA-Next/OneVision, Qwen2-VL, InternVL2, NVLM, etc.

---

> ### Author Response · Authors · 2024-11-25
> **Reponse to Reviewer Zi9g (Part 1/2)**
>
> Thanks for your thoughtful feedback and insightful suggestions. We have carefully considered each of your suggestions and addressed them as follows：
>
> **Q1 About the novelty**
>
> > My main concern is the limited novelty. As the authors introduced in Sec.3, introducing psychometrics into AI is not a new idea, and the proposed metrics (Test-retest Reliability, Parallel-forms Reliability, Criterion Validity) are also proposed by previous works. So the evaluation part seems to evaluate the current benchmarks with an off-the-shelf idea, with limited novelty.
>
> Introducing psychometrics into AI is an emerging trend, with only a few works [1, 2, 3] exploring this integration, each focusing on different aspects compared to our work. Specifically, [1] and [2] evaluate the psychological traits of LLMs using psychometric inventories and datasets, while [3] introduces the concept of construct-oriented evaluation from psychometrics into AI evaluation.
>
> Our work uniquely combines psychometrics with hallucination benchmarks for LVLMs,  to our knowledge, proposing the first quality measurement framework for AI benchmarks. We adapt and redefine psychometric metrics for AI benchmarks, forming a clearly defined and practical framework.
>
> **Q2 About the general applicability and relevance to hallucination**
>
> > Following 1, the evaluation is a general idea that is unrelated to the Hallucination problem. We could apply it to other multi-modal benchmarks and have some similar conclusions. So I think it hardly provides some insight about the following study about hallucination.
>
> As discussed in the Section 5, the principles underlying our benchmark quality measurement are general and can be extended to other multi-modal benchmarks, which we consider an advantage. Our focus on hallucination benchmarks stems from the fact that hallucination is currently a significant and critical issue for LVLMs. Additionally, hallucination benchmarks are highly diverse, encompassing various tasks and metrics, and their evaluation results are often inconsistent, making it difficult to determine which are more credible.
>
> Through our quality measurement framework, we identify several specific issues with existing hallucination benchmarks. For example, in free-form benchmarks, the GPT-based hallucination scores lack sufficient accuracy and stability, while in caption benchmarks, the metrics are highly influenced by response length. We hope that our analysis can provide insights for improving and designing future hallucination benchmarks.
>
>
>
> **Q3 About the difference with common capability evaluation benchmark**
>
> > Similarly, the proposed HQHBench also seems like a common MLLM capability evaluation benchmark (for example, a free-form version of MMBench or SeedBench), rather than a hallucination evaluation benchmark. If we treat all the wrong answers as hallucinations, we should not view hallucination as an independent research topic.
>
> Hallucination evaluation and common capability evaluation benchmarks may share some overlapping perception tasks but  their evaluation focus and design principles differ fundamentally. Hallucination evaluation emphasizes depth, employing fine-grained perception tasks to further analyze specific deficiencies across different hallucination types (e.g., attribution, existence) . In contrast, common capability evaluation benchmarks (e.g., MMBench, SeedBench) prioritize breadth, comprehensively assessing a model’s performance across perception and cognition tasks including logical reasoning, and numerical computation. While there is some overlap in perception tasks, they are not identical. Our HQH focuses exclusively on tasks that reflect cross-modal consistency, excluding others such as facial recognition or image segmentation, which fall outside the scope of hallucination evaluation.
>
> The core of hallucination evaluation lies in determining whether a model's textual output aligns with its visual input, rather than merely judging the correctness of answers. HQH employs a free-form design to avoid the influence of potential response bias on evaluation results. For example, in closed-ended tasks, incorrect answers may result not only from hallucination but also from response bias, such as a tendency to answer "yes" or favor certain options. HQH evaluation focuses specifically on hallucination issues, providing targeted feedback for improvement, while common capability evaluation holistically assess a model's overall abilities across a wide range of tasks, which are complementary, addressing different evaluation goals.

---

> ### Author Response · Authors · 2024-11-25
> **Reponse to Reviewer Zi9g (Part 2/2)**
>
> **Q4 About the capacity of GPT-3.5**
>
> > Last, I have a concern about the evaluation quality of HQHBench, the GPT-3.5 seems not capable enough to understand the complex image with only a short caption and phrases with bbox.
>
> In our criterion validity measurement, we examine the correlation between HQH evaluation using GPT-3.5 and human evaluation. The results show a correlation of 0.94, indicating that the evaluation quality of GPT-3.5 is close to human-level performance.
>
>
>
> **Q5 About the evaluation of new models**
>
> > Besides, the MLLMs evaluated in the paper are quite out-of-date. There are many capable new models, such as LLaVA-Next/OneVision, Qwen2-VL, InternVL2, NVLM, etc.
>
> Thank you for your suggestion. We have updated our evaluation on HQH to include several latest models. The results are presented in Table 1, showing that some of these new models perform exceptionally well, particularly GLM-4V and Qwen2-VL, with GLM-4V even surpassing GPT-4o. These results will be included in the revised manuscript.
>
> **Table1 Evaluation results of new models on HQH**
>
> | Models                 | Attribute | Action | Counting | Environment | Comparison | Relation | OCR   | Existence | Overall |
> | ---------------------- | --------- | ------ | -------- | ----------- | ---------- | -------- | ----- | --------- | ------- |
> | InterVL2               | 0.253     | 0.332  | 0.446    | 0.328       | 0.394      | 0.454    | 0.400 | 0.644     | 0.406   |
> | LLaVA-OneVision        | 0.134     | 0.348  | 0.168    | 0.204       | 0.390      | 0.436    | 0.252 | 0.366     | 0.287   |
> | Qwen2-VL               | 0.134     | 0.080  | 0.150    | 0.230       | 0.176      | 0.228    | 0.220 | 0.344     | 0.195   |
> | GLM-4V                 | 0.098     | 0.068  | 0.158    | 0.100       | 0.162      | 0.306    | 0.154 | 0.286     | 0.167   |
> | InternLM-XComposer2-VL | 0.118     | 0.110  | 0.174    | 0.120       | 0.251      | 0.350    | 0.226 | 0.528     | 0.235   |
> | MiniCPM-Llama3-v2.5    | 0.138     | 0.126  | 0.198    | 0.250       | 0.238      | 0.316    | 0.210 | 0.646     | 0.265   |
> | mPLUG-Owl2             | 0.268     | 0.296  | 0.432    | 0.332       | 0.455      | 0.554    | 0.504 | 0.598     | 0.430   |
> | Phi-3-Vision           | 0.162     | 0.134  | 0.218    | 0.172       | 0.315      | 0.378    | 0.222 | 0.384     | 0.248   |
> | Yi-VL                  | 0.258     | 0.382  | 0.482    | 0.438       | 0.531      | 0.528    | 0.570 | 0.670     | 0.482   |
>
> **Reference**
>
> [1] Pellert et al. AI psychometrics: Using psychometric inventories to obtain psychological profiles of large language models. OSF preprint 2023.
>
> [2] Li et al. Quantifying AI psychology: A psychometrics benchmark for large language models. arXiv preprint 2024.
>
> [3] Wang et al. Evaluating general-purpose AI with psychometrics. arXiv preprint 2023.

---

> > ### Comment · Reviewer_Zi9g · 2024-11-29
> >
> > Thanks for the detailed rebuttal, but I hardly agree with the authors' opinions about Q2 and Q3.
> > For Q5, the performance of InternVL2 is quite strange, as it is a leading LVLM while performing much worse than other concurrent models. Any idea about this?
> > Based on the above problem, I would keep my score as 5.

---

> > > ### Author Response · Authors · 2024-11-29
> > > **Reponse to Reviewer Zi9g**
> > >
> > > Thanks for your feedback. We fully understand your viewpoints regarding Q2 and Q3, and we still want to take an opportunity to further explain our work.
> > >
> > > Regarding Q2, we acknowledge your point that our quality measurement is a general idea. However, we are the first to evaluate the quality of hallucination benchmarks, as pointed out by reviewer NrvV and yourself, which we believe still provides a relatively new perspective on hallucination evaluation. We will also further refine this framework to achieve broader applicability based on your valuable feedback.
> > >
> > > Regarding Q3, hallucination evaluation primarily focuses on perceptual errors caused by modality misalignment, rather than addressing other traditional perception tasks involved in common capability evaluations (e.g. face recognition). In our work, we follow GAVIE and MMHal and concentrate on eight tasks closely related to hallucination. In evaluation criteria, hallucination evaluation places particular emphasis on the consistency between model's textual output and the visual input, rather than focusing on the correctness of answer as in capability evaluation. Even if the model’s answer is correct, we still consider it contains hallucination if any inconsistency arises in the generated explanation or analysis.
> > >
> > > Regarding Q5, we apologize for not providing a clearer statement earlier. Due to the time constraints during the rebuttal period (before extension) and limited computational resources, the initial evaluation was conducted using InternVL2-2b, which explains the relatively lower performance. We are currently conducting evaluation of InternVL2-8b and will promptly report the new results before the rebuttal deadline.
> > >
> > > We greatly appreciate your valuable insights, and we will continue to improve and refine our work.

---

> ### Author Response · Authors · 2024-11-30
> **Updated Evaluation Results**
>
> We have updated our evaluation results with InternVL2-8b as follows:
>
> **Table1 Evaluation results of new models on HQH**
>
> | Models                    | Attribute | Action | Counting | Environment | Comparison | Relation | OCR   | Existence | Overall |
> | ------------------------- | --------- | ------ | -------- | ----------- | ---------- | -------- | ----- | --------- | ------- |
> | InterVL2-2b               | 0.253     | 0.332  | 0.446    | 0.328       | 0.394      | 0.454    | 0.400 | 0.644     | 0.406   |
> | InterVL2-8b               | 0.188     | 0.226  | 0.312    | 0.234       | 0.257      | 0.358    | 0.326 | 0.240     | 0.268   |
> | LLaVA-OneVision-Qwen2-7b  | 0.134     | 0.348  | 0.168    | 0.204       | 0.390      | 0.436    | 0.252 | 0.366     | 0.287   |
> | Qwen2-VL-7b               | 0.134     | 0.080  | 0.150    | 0.230       | 0.176      | 0.228    | 0.220 | 0.344     | 0.195   |
> | GLM-4V-9b                 | 0.098     | 0.068  | 0.158    | 0.100       | 0.162      | 0.306    | 0.154 | 0.286     | 0.167   |
> | InternLM-XComposer2-VL-7b | 0.118     | 0.110  | 0.174    | 0.120       | 0.251      | 0.350    | 0.226 | 0.528     | 0.235   |
> | MiniCPM-Llama3-v2.5-8b    | 0.138     | 0.126  | 0.198    | 0.250       | 0.238      | 0.316    | 0.210 | 0.646     | 0.265   |
> | mPLUG-Owl2-Llama2-7b      | 0.268     | 0.296  | 0.432    | 0.332       | 0.455      | 0.554    | 0.504 | 0.598     | 0.430   |
> | Phi-3-Vision              | 0.162     | 0.134  | 0.218    | 0.172       | 0.315      | 0.378    | 0.222 | 0.384     | 0.248   |
> | Yi-VL-6b                  | 0.258     | 0.382  | 0.482    | 0.438       | 0.531      | 0.528    | 0.570 | 0.670     | 0.482   |
>
> We will continue to update our evaluation results along with the development of LVLMs.

---

### Official Review · Reviewer_3cYQ · 2024-11-01

**Soundness:** 3
**Presentation:** 3
**Contribution:** 3
**Rating:** 6
**Confidence:** 4

**Summary:**

This work proposes to look at multi-modal hallucination benchmarking from a novel perspective, i.e. the property of existing benchmarks. Further, on top of these properties, the paper presents a newly curated hallucination benchmark, providing more reliable and trustworthy evaluation of LVLMs.

**Strengths:**

- The work systematically studies the reliability and robustness for existing LVLM benchmarks. It provides quantitative evidences, characterizing various benchmarks' weakness in reliability (yes-and-no) and validity (free-form).
- The curated HQHBench effectively leverage existing annotations from Visual Genome, and have demonstrated that such a benchmark provides both reliable and valid signal for LVLM evaluation.

I also appreciate that the authors also managed to provide an anonymous data hosting the actual benchmark, which greatly increase the credibility of this work and the speed of the community to benefit from it.

**Weaknesses:**

One challenge I have is that some efforts of (in-)validating existing benchmarks are already creating new benchmarks that come with certain properties:
- we can combine POPE and "POPE-rewrite" as a benchmark that is robust against this parallel-form check, because it has been saturated with parallel forms.
- By utilizing the latest models, like GPT4o/o1, Claude 3.5 etc, validity may improve as well.

Essentially, the effort of curating a new benchmark is orthogonal to improving existing benchmarks. Even more I'd say they are additive to some extend, as this is a more efficient way to merge community efforts for a more comprehensive benchmark. I'd like to see authors' discussion on this point.

Another issue is that the HQHBench is curated with the bless of Visual Genome annotation, therefore hallucination judgement can be done in a binary format against image information provided in the annotation. However, there are still image information not covered by the annotation. Such a missing-annotation scenario may cause a rich response capturing not-in-annotation fact be judged as hallucination. How can we address such scenarios?

**Questions:**

- The test-retest protocol is a bit unclear. When we re-run the benchmark, do we require the model being evaluated to re-generate the responses? If not, why is there difference for Yes-or-No benchmarks? If yes, how do we distinguish whether the randomness comes from the benchmarking judge, or the models being evaluated?
- Visual Genome dataset's annotation is known to be noisy. How does this benchmark mitigate this issue?
- Can you provide an example of what POPE has been re-written into? Since POPE itself contains random/adversarial settings that prevents a degraded all-Yes model to pass, it already introduces parallel forms.

---

> ### Author Response · Authors · 2024-11-25
> **Reponse to Reviewer 3cYQ (Part 1/2)**
>
> Thanks for your thoughtful feedback and insightful suggestions. We have carefully considered each of your suggestions and addressed them as follows：
>
> **Q1 About the discussion on benchmark improvement and integration**
>
> > Essentially, the effort of curating a new benchmark is orthogonal to improving existing benchmarks. Even more I'd say they are additive to some extend, as this is a more efficient way to merge community efforts for a more comprehensive benchmark. I'd like to see authors' discussion on this point.
>
> Thank you for your insightful suggestions. The two points you raised are indeed valid and worthy of further exploration. Our core contribution lies in introducing the concept of quality measurement. We believe that both existing and future benchmarks can benefit from improvements based on our HQM framework, which serves as one of the primary goals of our work. By incorporating the perspective of quality measurement, we aim to provide new insights and references for the enhancement and design of hallucination benchmarks.
>
> Regarding your point about integrating improved benchmarks to form a more comprehensive and unified benchmark, we fully agree with its value. This approach not only facilitates the efficient consolidation of community efforts but also contributes to establishing a more valid and reliable benchmarking system. We plan to explore this direction further in the future and look forward to collaborating with the community to achieve this goal.
>
>
>
> **Q2 About the potential missing-annotation scenarios**
>
> > Another issue is that the HQHBench is curated with the bless of Visual Genome annotation, therefore hallucination judgement can be done in a binary format against image information provided in the annotation. However, there are still image information not covered by the annotation. Such a missing-annotation scenario may cause a rich response capturing not-in-annotation fact be judged as hallucination. How can we address such scenarios?
>
> While potential missing-annotation scenarios may exist, their impact on overall evaluation results is minimal. On one hand, our questions are designed to focus on specific aspects, such as the attributes of a particular object, rather than broad image captioning tasks, which constrains the model's responses, reducing the likelihood of overly divergent answers. On the other hand, the image information we utilize is highly comprehensive, covering nearly all major objects within the images.
>
> Additionally, in our validity measurement, we compare the hallucination evaluation conducted by human evaluators based on visual image with those by GPT using textual annotations. The results show a strong correlation, indicating that potential missing-annotation scenarios are minimal and have limited impact on the overall evaluation results.
>
>
>
> **Q3 About the details of  test-retest protocol**
>
> > The test-retest protocol is a bit unclear. When we re-run the benchmark, do we require the model being evaluated to re-generate the responses? If not, why is there difference for Yes-or-No benchmarks? If yes, how do we distinguish whether the randomness comes from the benchmarking judge, or the models being evaluated?
>
> Under the test-retest protocol, the evaluated model is required to regenerate its responses. For Yes-or-No and caption Benchmarks, the randomness originates from the models themselves. For free-form Benchmarks (GAVIE and MMHal) , randomness arises from both the models and GPT scoring. We believe that the introduction of GPT randomness leads to the relatively lower test-retest reliability. To address this, we improve the GPT evaluation process by modifying the instable scoring tasks to more reliable judgment tasks, reducing the impact of GPT randomness on evaluation results.
>
> To validate our analysis of randomness, we conduct repeated tests on the same model responses for MMHal and GAVIE, isolating GPT randomness, and calculate the correlation between the results. The correlation coefficients are 0.89 for MMHal and 0.88 for GAVIE, confirming that the randomness of GPT scoring is the primary source of their inconsistency results in repeated tests.

---

> > ### Author Response · Authors · 2024-11-25
> > **Reponse to Reviewer 3cYQ (Part 2/2)**
> >
> > **Q4 About the potential annotation noise**
> >
> > > Visual Genome dataset's annotation is known to be noisy. How does this benchmark mitigate this issue?
> >
> > Although the Visual Genome dataset's annotations are generated by Amazon Mechanical Turk (AMT) workers following strict guidelines and are generally of high quality, being human-generated data, it may still contain minor textual noise, such as non-alphabetic characters, stopwords, or occasional spelling errors.
> >
> > To mitigate this issue, on one hand, we manually reviewed all image-instruction pairs, removing samples affected by noise, such as those with incorrect ground truth answers. On the other hand, our evaluation criteria define "non-hallucination" based on the semantic similarity between the LVLM response and the ground truth annotation, which ensures that minor textual noise does not affect GPT judgment.
> >
> > Additionally, our validity measurement results shows a high correlation between GPT evaluation based on the textual annotations and human evaluation based on visual image, which further confirms the reliability of our evaluation based on these annotations.
> >
> >
> >
> > **Q5 About the re-writing**
> >
> > > Can you provide an example of what POPE has been re-written into? Since POPE itself contains random/adversarial settings that prevents a degraded all-Yes model to pass, it already introduces parallel forms.
> >
> > We provide an example of re-writing POPE as follows: "Is there a tree in the image?" → "Is there no tree in the image?"
> >
> > Using such re-written parallel forms, we can identify noticeable response biases in some models when answering Yes-or-No questions. For instance, in addition to the mentioned all-Yes models, which tend to answer 'Yes' to both forms, there are also all-No models that consistently answer 'No' regardless of the question's form. While POPE's random/adversarial setting can prevent all-Yes or all-No models from passing, it cannot distinguish whether a model's low performance is due to response bias or an inherent lack of capability in handling hallucinations. Our re-written parallel forms could help uncover such nuances.

---

> > > ### Comment · Reviewer_3cYQ · 2024-11-27
> > >
> > > Thanks for the detailed response, they address my concerns. I keep my score as recommendation, also would encourage these discussions will be incorporated in the final version of the paper for clarity.

---

> > > > ### Author Response · Authors · 2024-11-27
> > > > **Reponse to Reviewer 3cYQ**
> > > >
> > > > Thanks again for your valuable time and constructive feedback. We will incorporate these discussions into the revised manuscript to enhance its clarity and comprehensiveness. We sincerely appreciate your support for our work!

---

### Official Review · Reviewer_6UGx · 2024-11-04

**Soundness:** 3
**Presentation:** 3
**Contribution:** 2
**Rating:** 5
**Confidence:** 5

**Summary:**

This paper rethinks the reliability and validity of existing hallucination evaluation benchmarks, and therefore proposes test-retest and parallel testing methods for quantitative evaluation. Besides, this paper also curates a High-Quality Hallucination Benchmark (HQH) for comprehensively evaluating existing LVLMs hallucination degree.

**Strengths:**

Strengthens
1.	This paper investigates the instability of the evaluation metric of existing hallucination benchmarks, and contribute a new high-quality benchmark for the community.
2.	The writing quality is good and easy to follow.

**Weaknesses:**

1. The overall technical contribution is limited. The primary contributions of this paper include that existing evaluation metrics of hallucination benchmarks are instable under multi-fold validation, and curates a high-quality hallucination benchmark with “hallucination rate” as the evaluation metric.
2. The motivation and method are inconsistent. The major motivation of this paper lies in that the evaluation metrics of existing hallucination benchmarks are insufficient. But the proposed solution is to curating a new high-quality benchmark and calculate “hallucination rate” by altering the GPT-scores and GPT-based binary choices. The proposed multi-fold validation criterions are not employed for HQH evaluation.
3. The details of proposed designs are unclear. First, is it reasonable to use the validation set of Visual Genome dataset? It can incurs data leakage problem as existing LVLMs are using validation sets of open-source datasets for their training. Under this circumstance, test set is more suitable for establishing a benchmark. Second, what are the key differences between asking a LLM to directly give a score and judging whether the answer includes hallucination? I cannot figure out their intrinsic differences here.

**Questions:**

1. I'm wondering whether it is reasonable to use the validation set of Visual Genome dataset. It can incurs data leakage problem as existing LVLMs are using validation sets of open-source datasets for their training.
2. What are the key differences between asking a LLM to directly give a score and judging whether the answer includes hallucination? I cannot figure out their intrinsic differences here.

---

> ### Author Response · Authors · 2024-11-25
> **Reponse to Reviewer 6UGx (Part 1/2)**
>
> Thanks for your thoughtful feedback and insightful suggestions. We have carefully considered each of your suggestions and addressed them as follows：
>
> **Q1 About the motivation**
>
> > The motivation and method are inconsistent. The major motivation of this paper lies in that the evaluation metrics of existing hallucination benchmarks are insufficient. But the proposed solution is to curating a new high-quality benchmark and calculate “hallucination rate” by altering the GPT-scores and GPT-based binary choices. The proposed multi-fold validation criterions are not employed for HQH evaluation.
>
> Our primary motivation stems from the observation that existing hallucination benchmarks yield inconsistent evaluation results, making it challenging to determine which results are more trustworthy. To address this issue, we introduce a quality measurement framework aimed at quantifying the reliability and validity of hallucination benchmarks. Based on our analysis of the quality measurement results of existing benchmarks, we improve the metric for open-ended VQA and propose our HQH. We further validate that, under the quality measurement framework, i.e.  the mentioned "multi-fold validation criterions," HQH provides more reliable and valid hallucination evaluation.
>
>
>
> **Q2 About the potential data leakage**
>
> > I'm wondering whether it is reasonable to use the validation set of Visual Genome dataset. It can incurs data leakage problem as existing LVLMs are using validation sets of open-source datasets for their training.
>
> Thanks for raising this concern. To assess the potential risk of data leakage, we apply the Multimodal Leakage (ML) metric [1], which is designed to quantify the extent of data leakage in multimodal benchmarks. Specifically, ML calculates the difference in scores between an LVLM without visual inputs and its LLM base (without multimodal training) under the given benchmark. Higher ML value indicates more potential data leakage, as it suggests that the model performance without visual input surpasses that of its unimodal base, likely due to exposure to evaluation samples during multimodal training. Conversely, an ML value close to 0 indicates no data leakage.
>
> We calculate ML for the top-performing models on our HQH benchmark, as shown in Table 1. For comparison, we include ML of other benchmarks as reported in [1]. The results show that HQH achieves the lowest average ML across models compared to other benchmarks, demonstrating that HQH has minimal data leakage.
>
> Additionally, our HQH is relatively challenging, as most models perform poorly, which further supports that our benchmark effectively differentiates models and avoids inflating performance due to potential data leakage.
>
> **Table1 Multimodal Leakage↓ (%) of LVLMs on HQH and other benchmarks**
>
> | Models             | HQH     | SEEDBench | MMBench | ScienceQA |
> | ------------------ | ------- | --------- | ------- | --------- |
> | LLaVA-1.5-13b      | 1.0     | 10.7      | 9.8     | 7.0       |
> | LLaVA-1.5-7b       | 0.8     | 4.9       | 9.2     | 5.2       |
> | Qwen-VL            | 2.3     | 11.9      | 0.3     | 4.0       |
> | Gemini-1.5-Pro\*     | 1.5     | 0.0       | 0.0     | 0.0       |
> | GPT-4o\*             | 0.7     | 18.3      | 5.4     | 3.9       |
> | Average | **0.5** | 5.4       | 3.8     | 1.7       |
>
> \*For closed-source LVLMs, we compare the results with those of GPT-4V and Gemini-Vision-Pro as reported in [1].

---

> > ### Comment · Reviewer_6UGx · 2024-12-03
> > **Re-authors' response**
> >
> > Thank you for the authors' response. While my Q2 has been addressed, I still find it difficult to establish a strong association between the "quality measurement framework" and HQH, as they appear to be relatively independent of each other. Regarding Q3, I agree that binary classification is simpler than continuous scoring, but this introduces a new issue:
> > Given two answers, A1 and A2, where A1 contains 10 mistakes while A2 has only 1, the binary classification method would treat both as equally incorrect. In contrast, scoring judgment can distinguish the varying degrees of hallucination between the two.

---

> ### Author Response · Authors · 2024-11-25
> **Reponse to Reviewer 6UGx (Part 2/2)**
>
> **Q3 About the difference of GPT scoring and judgement**
>
> > What are the key differences between asking a LLM to directly give a score and judging whether the answer includes hallucination? I cannot figure out their intrinsic differences here.
>
> The key difference lies in the validity and reliability of the two metrics. Direct scoring requires LLMs to assign specific hallucination scores to LVLM responses, which current models often struggle to do accurately and consistently, due to their limited capability for precise quantification. In contrast, judging whether an LVLM response includes hallucination is a binary task, which is comparatively simpler and aligns better with the abilities of current LLMs.
>
> In our criterion validity measurement, we calculate the correlations between human evaluation and both GPT scoring and GPT judgment separately. The results show that GPT judgment achieves a higher correlation with human evaluation, indicating that it provides a more accurate and reliable results.
>
>
>
> **Q4 About the contribution**
>
> > The overall technical contribution is limited. The primary contributions of this paper include that existing evaluation metrics of hallucination benchmarks are instable under multi-fold validation, and curates a high-quality hallucination benchmark with “hallucination rate” as the evaluation metric.
>
> Our core contribution lies in introducing a hallucination benchmark quality measurement framework and proposing an improved benchmark. To the best of our knowledge, our HQM is the first framework aimed at measuring the quality of AI benchmarks, which is an innovative integration of AI and psychometrics. We belief this framework will help to identify potential  reliability and validity issues in existing benchmarks, providing valuable insights for current benchmark improvement and future benchmark construction within the community.
>
>
>
> **Reference**
>
> [1] Chen et al. Are we on the right way for evaluating large vision-language models? NeurIPS 2024.

---

> ### Author Response · Authors · 2024-12-03
> **Looking Forward to Your Feedback**
>
> Dear Reviewer 6UGx,
>
> We sincerely appreciate your constructive suggestions and have carefully addressed each of them. As the discussion phase is coming to an end, we look forward to your feedback on our response and welcome any further discussion on points that may remain unclear regarding both our paper and our responses.
>
> Thanks for your continued support of our work.
>
> Best regards!
>
> The authors

---

> ### Author Response · Authors · 2024-12-03
> **Reponse to Reviewer 6UGx**
>
> Thanks for your follow-up feedback!
>
> Regarding the "quality measurement framework", it is inspired by psychometrics and generally designed to evaluate the quality of hallucination benchmarks, incorporating indicators for both reliability and validity. Using this framework, we have identified and analyzed some quality issues of existing hallucination benchmarks. Through our analysis, we develop HQH, which is verified to provide more stable and accurate evaluation results under this framework.
>
> Regarding Q3, we agree with your point that scoring can distinguish the degree of hallucination. However, the current issue is that LLMs, including GPT, still have limitations in achieving precise and consistent scoring, as evidenced by our experiments. We believe that for a metric to be effective, it is more crucial to ensure the accuracy and consistency of the evaluation results compared to granularity. Therefore, binary classification is a more practical choice at present. In the future work, we will continue to explore new metrics that reflect the degree of hallucination with finer granularity.
>
> Please let us know if you have any remaining concerns, or if you would consider updating your evaluation based on our response. We sincerely appreciate your support of our work!

---

> > ### Comment · Reviewer_6UGx · 2024-12-03
> > **Re-authors' response**
> >
> > Thanks for your prompt feedback. I hold my opinions and keep the original recommendation.

---

### Meta-Review · Area_Chair_LVmL · 2024-12-20

**Metareview:**

This paper revisits the reliability and validity of existing hallucination evaluation benchmarks for large vision-language models (LVLMs). It proposes a High-Quality Hallucination Benchmark (HQHBench) alongside test-retest and parallel testing methods for evaluation. The work is praised for systematically studying the weaknesses of current benchmarks, providing quantitative evidence of their instability, and curating a new benchmark that leverages Visual Genome annotations to deliver more reliable and valid evaluation signals. Strengths include the clarity of writing, the novel perspective on hallucination benchmarking, and the contribution of a curated dataset shared with the community. However, reviewers express concerns about the limited technical contribution, inconsistencies in the motivation and method, and potential data leakage issues due to the use of Visual Genome's validation set. They highlight challenges such as noisy annotations, missing information in the dataset, and a lack of discussion on how HQHBench complements or improves upon existing benchmarks like POPE. While the contributions are acknowledged, the paper's impact is seen as incremental and does not fully address core issues, leaving it marginally below the acceptance threshold.

**Additional Comments On Reviewer Discussion:**

The reviewers remain unconvinced on key issues. One reviewer notes the difficulty of establishing a strong connection between the proposed "quality measurement framework" and HQH, as they seem independent. Additionally, concerns are raised about the binary classification approach for hallucination evaluation, as it fails to capture varying degrees of hallucination severity compared to continuous scoring methods. Another reviewer questions the unexpectedly poor performance of InternVL2, a leading LVLM, and finds the authors' rebuttal insufficient to address this anomaly. Despite some clarifications, the concerns about methodological choices and unexplained results lead both reviewers to maintain their scores, indicating skepticism about some of the paper's contributions.

---

### Decision · Program_Chairs · 2025-01-22

Reject